# Modeling the Territorial Structure Dynamics of the Northern Part of the Volga-Akhtuba Floodplain

Inessa I. Isaeva [†] [ID], Alexander A. Voronin [†] [ID], Alexander V. Khoperskov [†] [ID] and Mikhail A. Kharitonov *,[†] [ID]

Volgograd State University, Universitetsky pr., 100, Volgograd 400062, Russia; isaeva-inessa@mail.ru (I.I.I.);
a.voronin@volsu.ru (A.A.V.); khoperskov@volsu.ru (A.V.K.)
* Correspondence: kharitonov@volsu.ru
† These authors contributed equally to this work.

**Abstract:** The subject of our study is the tendency to reduce the floodplain area of regulated rivers and its impact on the degradation of the socio-environmental systems in the floodplain. The aim of the work is to create a new approach to the analysis and forecasting of the multidimensional degradation processes of floodplain territories under the influence of natural and technogenic factors. This approach uses methods of hydrodynamic and geoinformation modeling, statistical analysis of observational data and results of high-performance computational experiments. The basis of our approach is the dynamics model of the complex structure of the floodplain. This structure combines the characteristics of the frequency ranges of flooding and the socio-environmental features of various sites (cadastral data of land use). Modeling of the hydrological regime is based on numerical shallow water models. The regression model of the technogenic dynamics of the riverbed allowed us to calculate corrections to the parameters of real floods that imitate the effect of this factor. This made it possible to use digital maps of the modern topography for hydrodynamic modeling and the construction of floods maps for past and future decades. The technological basis of our study is a set of algorithms and software, consisting of three modules. The data module includes, first of all, the cadastres of the territory of the Volga-Akhtuba floodplain (VAF, this floodplain is the interfluve of the Volga and Akhtuba rivers for the last 400 km before flowing into the Caspian Sea), satellite and natural observation data, spatial distributions of parameters of geoinformation and hydrodynamic models. The second module provides the construction of a multilayer digital model of the floodplain area, digital maps of floods and their aggregated characteristics. The third module calculates a complex territorial structure, criteria for the state of the environmental and socio-economic system (ESES) and a forecast of its changes. We have shown that the degradation of the ESES of the northern part of the VAF is caused by the negative dynamics of the hydrological structure of its territory, due to the technogenic influence the hydroelectric power station on the Volga riverbed. This dynamic manifests itself in a decrease in the stable flooded area and an increase in the unflooded and unstable flooded areas. An important result is the forecast of the complex territorial structure and criteria for the state of the interfluve until 2050.

**Keywords:** environmental and socio-economic system; territorial structure; hydrodynamic and geoinformation modeling; high performance computing; Volga-Akhtuba floodplain

## 1. Introduction

Floodplain ecosystems are extremely vulnerable landscapes. They are degraded due to violations of land-use regulations and changes in the characteristics of natural flood events that ensure the existence of such unique areas [1–5]. The state of river floodplains depends mainly on the nature of their flooding, whose hydrological regime is determined by the spring flood volume, a channel structure and a relief of the territory [1,6–12].

The ability to regulate the natural hydrological regime has always been the main factor for the creation of socio-economic systems in river valleys and, in particular, in floodplain

areas [3,12,13]. Increased deviation of hydrological characteristics from the natural regime leads to large losses in ecosystems. Even slow human-induced degradation of floodplain channel systems sooner or later becomes the cause of an imbalance in the hydrological state and functional significance of a noticeable part of such territory. This trend occurs in the ecological degradation of wetlands, a decrease in the productivity of fisheries, a drop in agricultural productivity and floodplain landscape steppification [14]. Such negative natural and anthropogenic dynamics are also a factor that complicates the ecological and economic management of the floodplain area due to the impossibility of identifying damage caused by the activities of specific economic entities [15].

The Yellow River and the Yangtze River floodplains are the most famous historical examples of China's largest water systems anthropogenic degradation [16,17]. Anthropogenic accumulation of river sediments in the lower reaches was the result of the dams' construction, which begun about 2900–2700 years ago for flood protection. The sediment accumulation required a constant increase in the number and height of dams, which only increased the rate of sediment inflow. The famous Nile floodplain and the Tigris and Euphrates interfluves are also classic examples of natural ecosystem degradation [1,18].

The construction of dams in the Amudarya River channel to accumulate water for agricultural and social needs in various periods of the 7th–16th centuries is a lesser-known example of a river system's anthropogenic degradation, which caused not only ecologic, but geoeconomic and even geochronological consequences. The result of such a systematic impact was a change in the river flow from the Caspian Sea to the Aral Sea, draining of the 550-km riverbed of the Uzboy River (part of the old AmuDarya channel system), the fertile land desertification in an area of more than 200,000 km$^2$ and the change in the route of the Silk Road from China to Europe, stagnating the centuries-old socio-economic situation of the peoples inhabiting this territory [19]. A single class of the floodplain territories is formed by the interfluves, when the floodplain is located between two close, approximately parallel rivers. The Tigris and Euphrates interfluve is an example of such territories. The active economic activity there was the cause of secondary soil salinization and sediment accumulation, which led to channel shallowing and divagation [18]. It is believed that these phenomena were the main reason for the desolation of many Mesopotamia territories at various periods of its history. The modern wetland complex in Iraq (Central or Qurna Marshes) is estimated at only 3000 km$^2$, including the Hawizeh and Hammar marshes as a part of the Tigris–Euphrates river system [18].

The main problem in the floodplains of unregulated rivers is flooding. On the contrary, the floodplains of regulated rivers suffer from aridization due to the hydrological regime, which is determined by the needs of hydropower and the safety of hydraulic structures [20]. The flood falls and the disappearance of regular floods lead to the river valleys drying up and, as a result, the wetland degradation with the transition to substitutive ecosystems. The flood hydrological regime regulation creates the possibility of active socio-economic development of the floodplain territories, which requires a significant amount of water resources and causes additional anthropogenic pressure on the ecosystem. Thus, the main problem of managing the floodplain areas of regulated rivers is to ensure the ecosystem sustainability and the sustainable socio-economic development in the context of water scarcity [2,9,21].

The sustainable development of floodplain territories is determined by the hydrological regime, which ensures both the sustainability of socio-economic development and the preservation of their ecosystems. Floodplain ecosystems are based on territories that are stable flooded in spring with frequencies exceeding 0.85 during the observation period. Unflooded areas of the floodplain are the basis of their socio-economic infrastructure. The rest of the floodplain (unstable flooded area) is primarily of recreational importance. Therefore, the trend towards a decrease in the stable flooded and growth of the unstable flooded areas under the influence of natural and technogenic factors is the main reason for their complex degradation.

The purpose of our study is to create a new approach to the analysis and forecasting of the trend of complex environmental and socio-economic degradation of floodplain territories under the influence of various factors. This approach uses a combination of methods of hydrodynamic and geoinformation modeling, statistical analysis of observational data and results of computational experiments. The basis of our approach is the dynamics model of the complex structure of the floodplain territory. This structure integrates the characteristics of the frequency ranges of floods and the socio-natural economic significance of various parts of the territory.

The regression model of the technogenic dynamics of the riverbed allows us to calculate corrections to the parameters of real floods that imitate the effect of this factor. This makes it possible to use digital maps of modern relief for hydrodynamic modeling and flood mapping for past and future decades. We find the minimum allowable sample size from the spring flood data array to determine the parameters of their distribution function with acceptable accuracy. An assessment of the variability of these parameters over the observation period underlies the accuracy of forecasting the territorial structure dynamics. The difference between the volume of observational data and the sample size determines the period of modeling of such dynamics. Thus, the condition for the feasibility of our approach is a noticeable excess of the volume of observational data over the sample size (in our study, it is equal to 2). We use the developed approach for the territory of the Volga-Akhtuba floodplain (Figure 1), which is affected by complex degradation factors. We hope that the results of applying such an approach can form the basis for designing adequate ways to ensure the sustainable development of other floodplains as well.

Each specific floodplain area requires an individual approach to modeling the hydrological regime and analyzing hydrological processes that determine the ecological state and economic significance of lands [2,3,9,12,17]. To measure this value, the concept of "ecosystem services" is used, which are characterized by useful functions of a given territory [22]. For regularly flooded areas, such functions depend on the following factors: rich vegetation cover, which is a protection against storm water runoff and water erosion, species diversity in the floodplain compared to the external environment, providing unique biogeochemical characteristics of the system, and humus and moisture content in the soil, which underlies the high efficiency of agriculture. It is important to note that biological species of both flora and fauna in the river floodplains are often represented by rare or even endemic species. Unique biological diversity influences cultural and educational services as a part of recreational resources. The integral value of all wetlands worldwide seems to exceed US\$5 trillion $yr^{-1}$, while the total area of the floodplain territories has decreased by about 10 times over the last century [23,24].

The Volga-Akhtuba floodplain (VAF) is a unique natural feature stretching for almost 450 km in the semi-desert zone of Southern Russia. It is practically the only long section of the Volga River that has largely retained its natural state (see Figure 1), and therefore, it occupies a special position among the interfluves of the world. The Proto valley of the Lower Volga River formed over the past 600–700 thousand years, periodically turning into a deep and long ingressive bay, whose northern boundary was determined by the rise in the Caspian Sea level. During the entire Late Pleistocene and Holocene, this bay alternated between liman-sea and river regimes due to fluctuations in the Caspian Sea level. The sea freed the valley during the regressive phase, and at that time, the floodplain was forming [25]. During the transgressive phases, the sea spread up the Volga valley and formed an ingressive bay. Such transitions had occurred six times within the Volga-Akhtuba valley over the past 16 thousand years. Sections of river floodplains formed in previous regressive stages were partially preserved in subsequent transgressive intervals and gradually increased the overall delta, which then turned into the modern Volga-Akhtuba floodplain (see insets in Figure 1). We model the hydrological regime of the northern part of the VAF only (see the light blue box on the top right of Figure 1). This section contains the source of the Akhtuba River, which is a key point for the hydrology of the entire VAF and has been the most affected by urbanization processes in recent years.

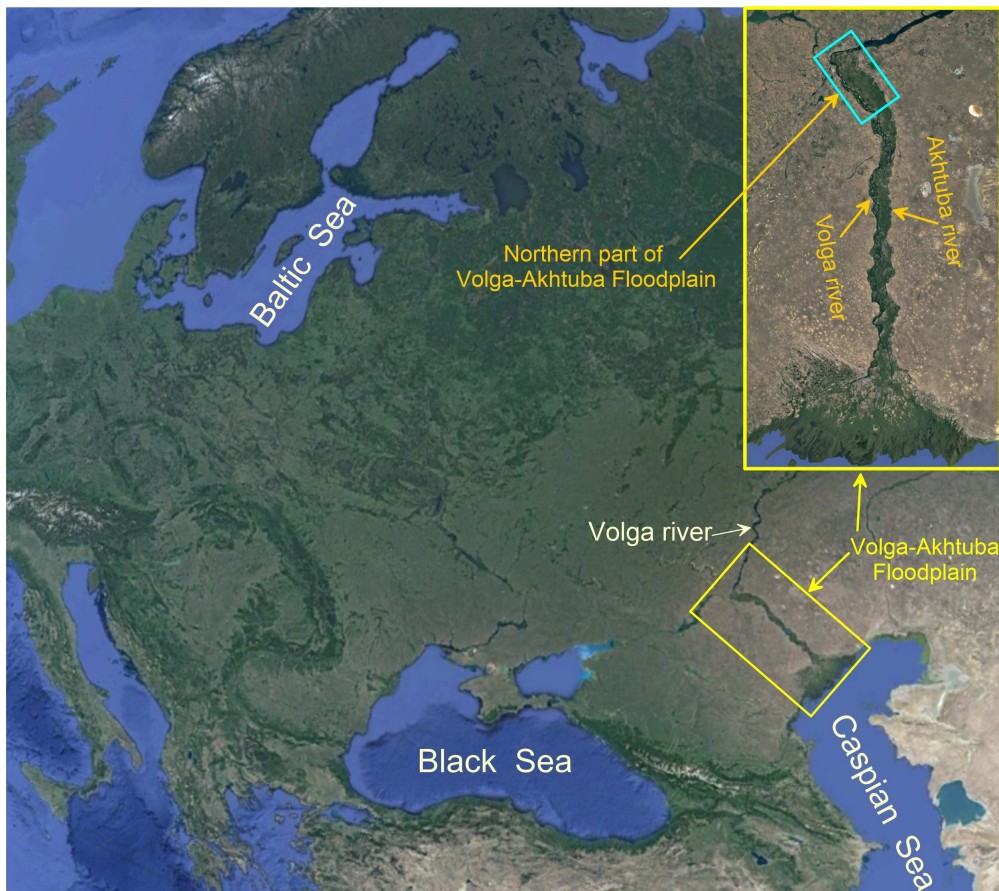

**Figure 1.** The Volga-Akhtuba floodplain is located in the Lower Volga region from Volgograd City to the Caspian Sea and the length of the interfluve is about 450 km (Google Map images).

The modern VAF is characterized by high natural diversity, the presence of rare and vulnerable plant and animal species under existing conditions, as well as favorable opportunities for the development of agriculture and ecological tourism. The VAF natural landscape is conditioned by the spring discharge of water, which floods a significant part of its low-lying areas. At present, the natural reproduction of sturgeons in the Volga has been significantly undermined. One of the most precious objects of protection of the VAF is the last remaining most productive natural sturgeon spawning grounds on the Volga with an area of only 250 hectares.

The creation of a cascade of the largest hydroelectric power stations and reservoirs on the Volga River launched the process of restructuring the riverbed and floodplain landscape. The functioning of the Volga Hydroelectric Power Station (VHPS) has become the main factor in the formation, functioning and degradation of the environmental and socio-economic system (ESES) of the VAF. The reduction in the average volume and peak values of floods after the VHPS launch stimulated the active socio-economic development of the vacated fertile wetlands. The reverse result of the positive socio-economic consequences of river regulation was a chain of negative factors that caused significant damage to the natural complexes of the VAF.

The form of flood hydrographs through the VHPS dam was changed at the turn of the 20th and 21st centuries. The domed function $Q(t)$ with numerous small oscillations (Figure 2a,b) has been replaced by a two-step form (Figure 2c). The type of discharge functions (or hydrographs) $Q(t)$ of different years demonstrates relative chaoticity throughout the 20th century, which is due to the safety requirements of the VHPS and variable consumer demand for electricity while ignoring environmental requirements. Since 2003, two stages have been distinguished with almost constant discharges $Q_1^{(m)}$ and $Q_2^{(m)}$, with

duration $\theta_1$ and $\theta_2$, respectively, (see Figure 2c). The flooding area of the interfluve in the case of a two-stage hydrograph is determined primarily by the value of $Q_1^{(m)}$ and the interval $\theta_1$ for the first stage, while the VAF flooding practically does not occur when $Q_1^{(m)}$ is less than 20,000 m$^3$ s$^{-1}$. The values of $Q_1^{(m)} > 30,000$ m$^3$ s$^{-1}$ at $\theta_1 > 5$ days correspond to catastrophic flooding for the existing infrastructure in the northern part of the VAF.

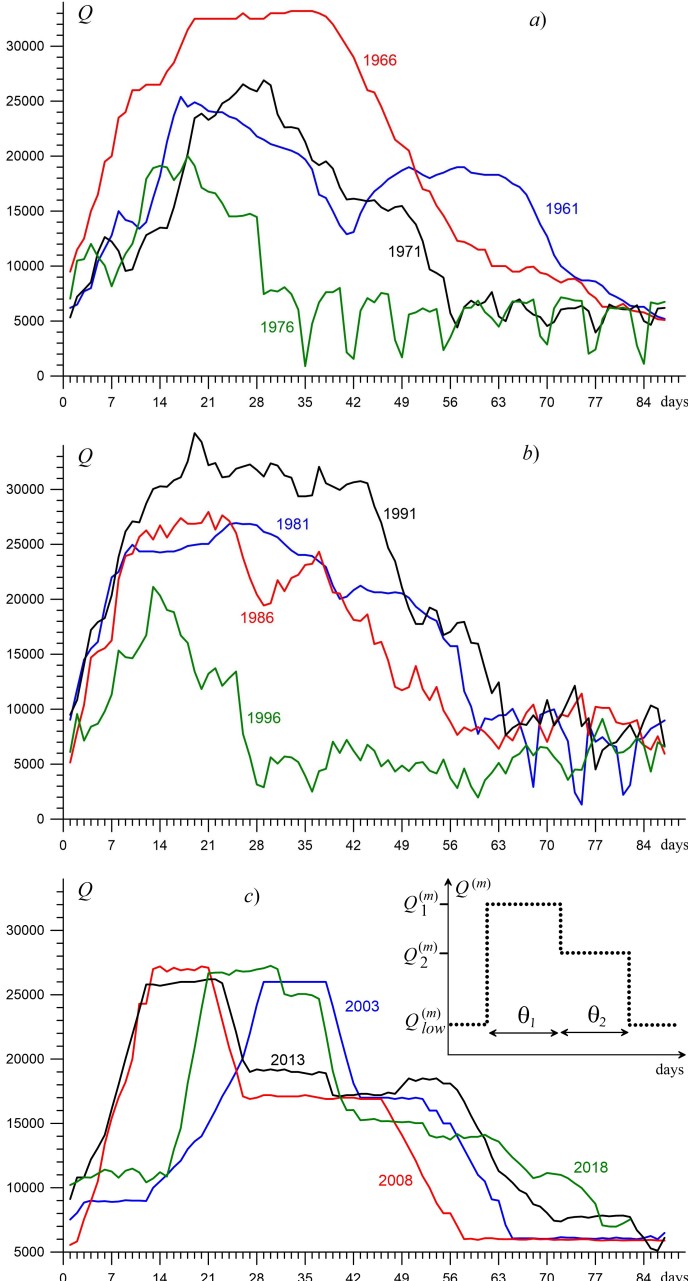

**Figure 2.** Examples of a flood discharge function (hydrograph) of the VHPS $Q(t)$ in different periods of its operation: (**a**) 1961–1980; (**b**) 1981–2000; (**c**) 2001–2021, inset shows a model of so-called two-stage hydrograph $Q^{(m)}(t)$, typical for the last 20 years. Such a model hydrograph is determined by a set of parameters for the following stages: (1) high constant water discharge $Q_1^{(m)}$ with duration $\Theta_1$ during spring flood; (2) lower level $Q_2^{(m)}$ with duration $\Theta_2$ during spring flood; (3) low water level $Q_{low}^{(m)}$.

After the VHPS construction, there was a significant increase in the share of winter and summer flows with a significant decrease in the maximum and average values of flood

peaks. The VHPS functioning has led to sediment transport disruption and changes in channel-forming processes. Therefore, the water level of the Volga River below the dam has decreased by about 1.3 m by now (according to the data from the gauging stations of the Volga River, http://gis.vodinfo.ru/, 26 February 2022). This fact has had an extremely negative effect on the flood water inflow into the Akhtuba River, which is the source of flood water for the most of the territory of the northern part of the VAF and source of water for small channels during the low-water period. This trend was exacerbated by climate change: winter warming (by 1.5–3.0 °C) with an increase in the number of thaws and the sum of positive temperature values, which has led to an increase in the share of rainfall, groundwater levels and surface winter inflow into rivers. These phenomena caused an additional increase in winter, summer and autumn discharge of water, and as a result, a decrease in the average volume of spring floods is observed. The results of climate modeling suggest that this trend will continue in the coming decades.

The desertification factors of the floodplain are both the "cutting off" of high peaks of water discharge during the high-water season (as a result, a decrease in flood self-cleaning of its channels), and the plowing of coastal zones and the forests area reduction near the channels. The floodplain drying is exacerbated by the construction of unauthorized earth-filled dams by local residents and business entities for the irrigation of fields and gardens in summer. These dams are one of the factors in reducing the average flood area in the VAF. Road construction has intensified the process of urbanization in recent decades. Infrastructural and urbanization factors reinforce each other, stimulating further reduction in flooding and an increase in the rate of degradation of the floodplain ecosystem and agricultural productivity.

The problem of water resources deficit formulated above is common for all regulated floodplains and is exacerbated for the interfluve of the VAF by the factors of progressive channel degradation and negative climate changes in recent decades. In this regard, the mathematical modeling of these phenomena is of particular relevance, which makes it possible to obtain fairly accurate quantitative estimates of negative impacts using high-performance computing, build a forecast for the development of the situation and on its basis design an adequate system for designing a territory development strategy (Section 3). Our control models are based on direct methods of hydrodynamic simulations of the flooding process (Section 2), which requires the construction of high quality geoinformation models to describe the spatial distributions of the physical characteristics of the study area. Solving the problems of optimizing the hydrological regime and managing the development of floodplain territories requires the use of large sets of spatial data (tens of thousands of digital flood maps for an area of several thousand square kilometers), and high-performance computing technologies make it possible to use effective methods for solving them.

## 2. Numerical Modeling of the Hydrological Regime of the Floodplain Area

The calculation of the dynamics of the large floodplain hydrological regime requires the use of high-quality tools for geoinformation and hydrodynamic modeling. We apply shallow water numerical models using high performance GPU codes [4,26–28]. The construction of updated digital terrain models, and above all, the digital elevation model is based on integrating data assimilation from various sources [29,30].

### 2.1. Geoinformation Modeling

Digital Elevation Model (DEM) is defined by ground elevation data, whose sources are a wide variety of remote sensing data, aerial surveys, including mapping drones, lidar usage, geodetic survey data, etc. The closely related term Digital Surface Model (DSM) is often used, taking into account buildings, vegetation, cultural and engineering structures. Many authors equate the terms of DEM and DTS [31]. In addition to vertical data, we use additional spatial matrices necessary for hydrodynamic modeling. Even a simple single-layer shallow water model, in addition to the height matrix $b(x, y)$, requires a sur-

face roughness matrix $n_M(x, y)$. More complex multilayer models with sediment and/or groundwater dynamics include additional matrices defining soil porosity $\psi(x, y)$, aquifer depth $b_g(x, y)$, sediment source and sink functions $q_w(x, y)$, coefficient of hydraulic conductivity $k_s(x, y)$, filtration coefficient $k_\phi(x, y)$, parameters $C_J$, $A_J$, $C_{Sh}$ defining properties of sedimentation [28]. We use the term "Digital Hydrological Landscape Model" (DHLM proposed by Anna Klikunova [30]), which includes the following set of spatial matrices:

$$\left\{ b_{ij}, n_{Mij}, \psi_{ij}, b_{gij}, q_{wij}, k_{sij}, k_{\phi ij}, C_{Jij}, \dots \right\},$$

where all quantities are calculated at the nodes of the given spatial grid ($x_i = i \cdot \Delta x, y_j = j \cdot \Delta y$) with a step $\Delta x = \Delta y$ within $\simeq$5–20 m, necessary for modeling the dynamics of surface water, groundwater and sediment with maximum computational domain size of approximately 40 km by 60 km (see the blue box in Figure 1) [4,27,28].

Figure 3 shows the main steps in the process of creating a digital elevation model. Assimilation of heterogeneous spatial data requires a sequential iterative procedure with verification at each stage. The basic DEM is based on the height matrix according to satellite data $b_{ij}^{[0]}$, which is refined and updated using various sources. Channel beds (the Volga River, the Akhtuba River and numerous small interfluve channels) are of critical importance. Our practice of constructing and using DEMs indicates the significant role of observational data on the dynamics of coastlines of flooded zones at various points in time, since such lines exactly coincide with the isolines of the height surface [30].

Figure 4 shows the DEM of the northern part of the VAF, where the Volga and Akhtuba channels and numerous small channels (the so-called eriks) are clearly visible. Typical absolute heights in this interfluve zone are within $(-1 \div -10)$ m. During the spring flood (see hydrographs in Figure 2), the water in the Volga River and the Akhtuba River rises by $6 \div 10$ m or even more, which ensures the flooding of a noticeable part of the flat territory between the rivers through the system small channels. It is important to emphasize that the main inundation occurs through the Akhtuba River, and the direct contribution from the Volga River is only 20–30% of the area for the northern part of the VAF.

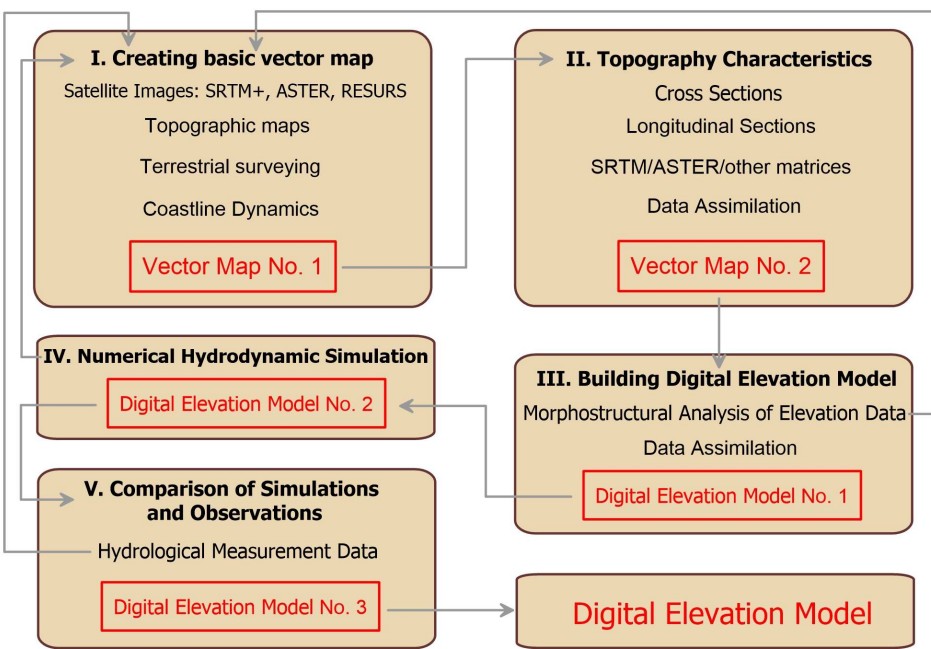

**Figure 3.** Diagram of creating and updating a digital elevation model [29,30].

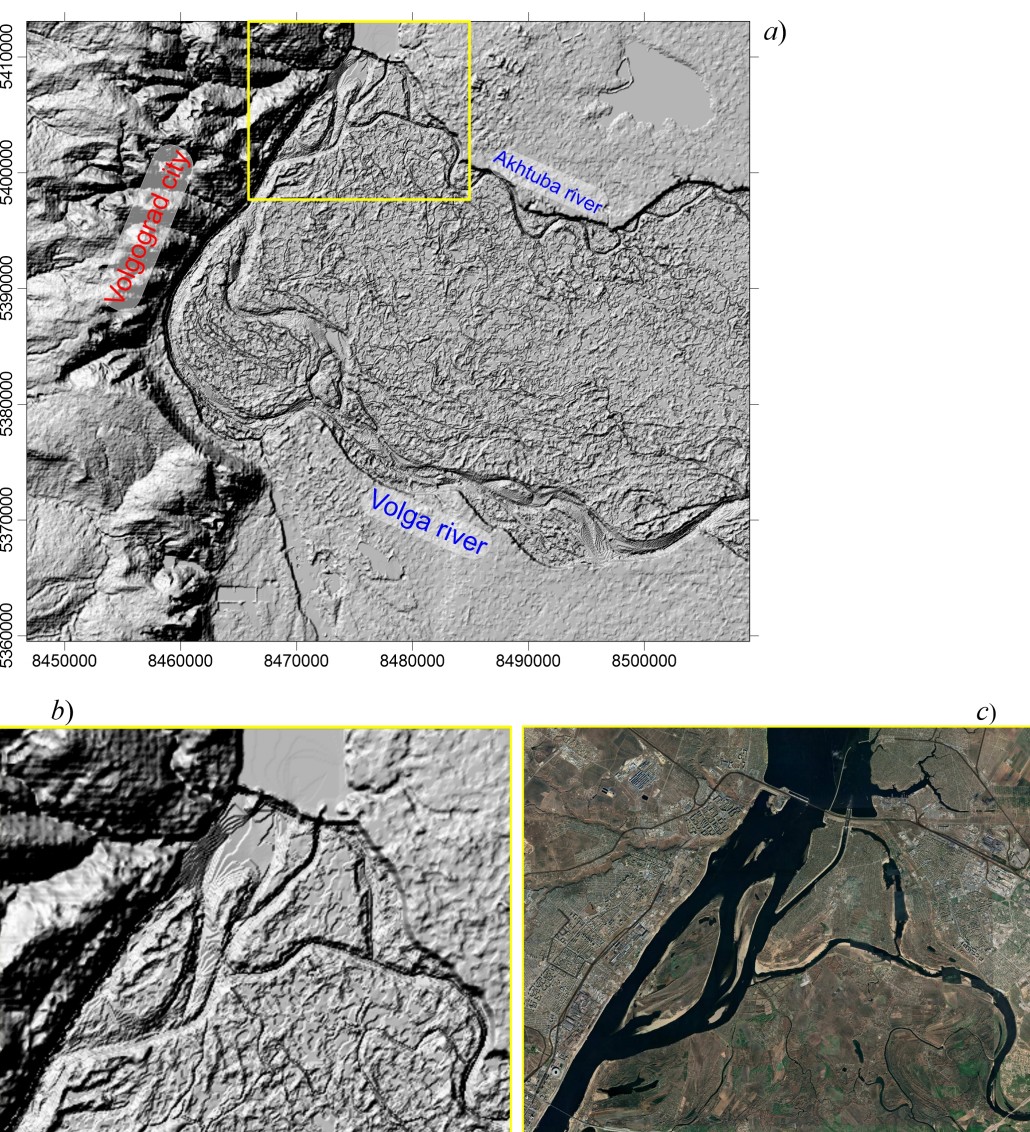

**Figure 4.** (**a**) DEM of the northern part of the VAF, 50 km by 60 km. Elevations are shown in the Shaded Relief Map model. Bottom: (**b**) DEM fragment highlighted with a yellow line on the main DEM (**a**,**c**) corresponding map of this zone (Google/Landsat-8 image). Long small channels are well traced in the interfluve of the Volga and Akhtuba rivers.

Figure 5 shows an example of spring flooding in the numerical model against the background of the DEM. The nature of the increase in water discharge $Q^{(m)}(t)$ through the VHPS dam in the spring (April–May) determines the hydrological regime throughout the whole interfluve [2,4,9]. The relief features do not allow flooding of the right bank of the Volga River and the left bank of the Akhtuba River. The main part of the water volume passes through the Volga River and only about 2–5% of $Q^{(m)}(t)$ branches off into the Akhtuba River and then only a part goes into the interfluve.

The parts of the DHLM are distributions of fluid sources that determine the operating modes of hydraulic structures, heavy rains, snowmelt and others. Figure 2 demonstrates the typical behavior of the hydrographs through the dam gate of the VHPS in different years, as calculated by:

$$Q(t) = \iint q_w^{(VHPS)}(x, y, t) \, dS \,, \tag{1}$$

where $q_w^{(VHPS)}$ is the surface density of water discharge through the dam. A characteristic feature for the 20th century is the unstructured form of the dependence $Q(t)$. For the

last 20 years, the so-called two-stage hydrograph has been used to improve the ecological situation in the floodplain, which has an approximately constant level of water inflow through the dam $Q = Q_1^{(m)} \simeq$ const at the maximum level (this stage is called "Agriculture shelve"). Then, a roughly constant flow $Q = Q_2^{(m)} =$ const $(Q_2^{(m)} < Q_1^{(m)})$ is maintained to conserve water in the interfluve, providing favorable conditions for fish fry, and the second stage is called "Fish shelve".

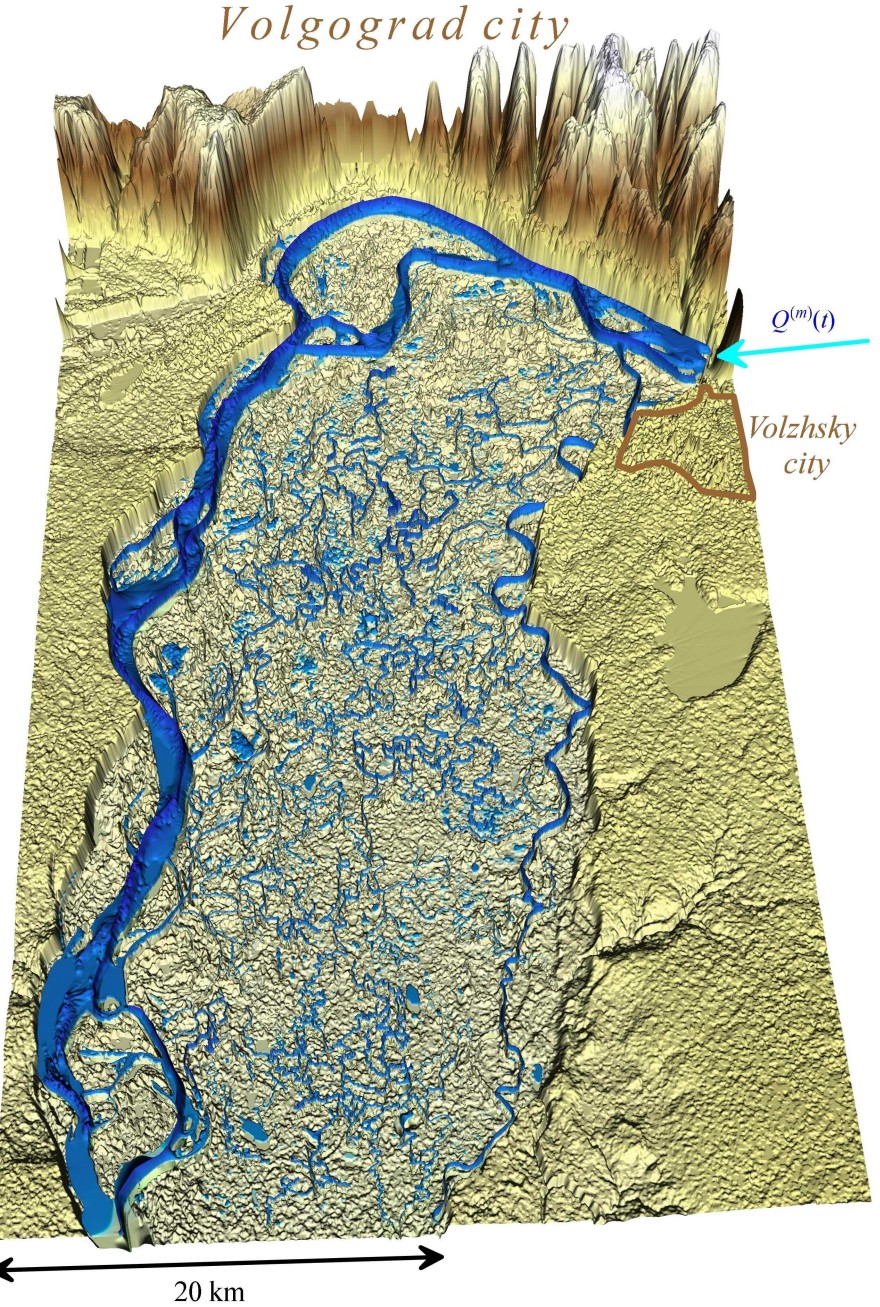

**Figure 5.** DEM of the northern part of the Volga-Akhtuba floodplain and its environs. The vertical scale differs from the plane-scale (the elevation level within the Volgograd city boundary is $50 \div 100$ m higher compared to the floodplain area). The blue color shows the water distribution from our numerical simulations.

The software for numerical modeling of the hydrological regime in the interfluve includes three main groups of components, as shown in Figure 6. The "Input Spatial Data Model" and "Processing and Visualization of Simulation Results" blocks are based on

geoinformation technologies that provide efficient operation with spatial-temporal data and geographic maps. The "Hydrodynamic Models" components compute the dynamics of hydrological processes (see Sections 2.2 and 2.3 below).

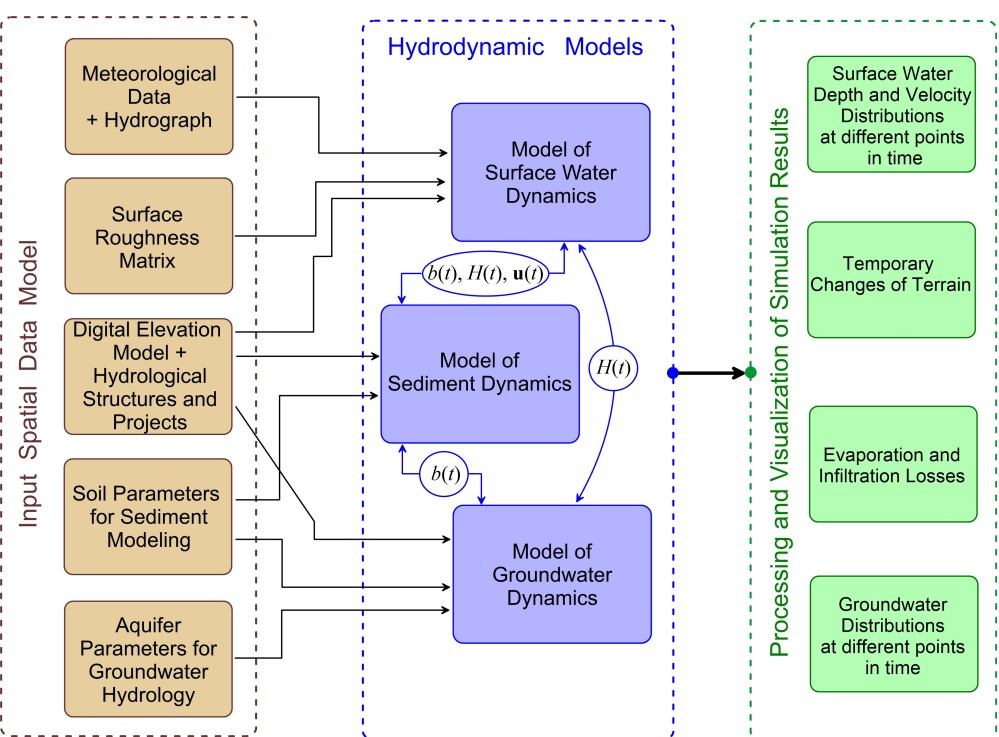

**Figure 6.** The main components of the computational model of the hydrological system.

### 2.2. Single-Layer Shallow Water Model

The shallow water model is the basis for modeling the dynamics of surface water over large areas, taking into account a realistic heterogeneous terrain [4,8,32–37]. Figure 7 shows the geometry of a one-layer shallow water model on a complex topography $b(x, y)$. The system of equations for a thin layer of an incompressible fluid is [4]:

$$\frac{\partial H}{\partial t} + \frac{\partial Hu}{\partial x} + \frac{\partial Hv}{\partial y} = q_w(x, y, t),$$ (2)

$$\frac{\partial Hu}{\partial t} + \frac{\partial Hu^2}{\partial x} + \frac{\partial Huv}{\partial y} = gH\frac{\partial(H + b)}{\partial x} + Hf_x,$$ (3)

$$\frac{\partial Hv}{\partial t} + \frac{\partial Huv}{\partial x} + \frac{\partial Hv^2}{\partial y} = gH\frac{\partial(H + b)}{\partial y} + Hf_y,$$ (4)

where the velocity components $\{u, vs.\} = \mathbf{u}$ are the average values along the vertical $z$-coordinate, $g$ is the acceleration of gravity, $\mathbf{f} = \{f_x, f_y\} = \mathbf{f}^{(diss)} + \mathbf{f}^{(wind)} + \mathbf{f}^{(sour)}$ is the force describing the dissipative factors ($\mathbf{f}^{(diss)}$), influence of wind ($\mathbf{f}^{(wind)}$) and momentum associated with the presence of fluid sources $q_w$ ($\mathbf{f}^{(sour)}$) and $q_w$ is the surface sources of water caused by the operation of hydraulic structures (see (1), precipitation, snowmelt, etc., including the contribution of rain runoff [38]. Equation (2) is the law of mass conservation in a thin liquid layer of variable thickness $H(x, y, t)$. Equations (3) and (4) are the $x$ and $y$ components of the motion law of an incompressible fluid layer on inhomogeneous surface $b(x, y)$ and are a special case of the Navier–Stokes equation for shallow water. The most

important factor is the dissipative forces $\mathbf{f}^{(diss)} = \mathbf{f}^{(turb)} + \mathbf{f}^{(M)}$, which are traditionally divided by the internal friction force:

$$\mathbf{f}^{(turb)} = \nabla_\perp(\nu_{turb}\nabla_\perp \mathbf{u}), \quad \nabla_\perp = \left\{\frac{\partial}{\partial x}, \frac{\partial}{\partial y}\right\}, \tag{5}$$

which is determined by the effective coefficient of turbulent viscosity $\nu_{turb}$ and the interaction of the flow with the underlying surface, for which various models are used, for example, the Manning model in the form [4,6,39–41]:

$$\mathbf{f}^{(M)} = -\frac{g n_M^2}{H^{4/3}} \mathbf{u} \cdot |\mathbf{u}|, \tag{6}$$

where $n_M$ is the Manning coefficient ($[n_M] = $s·m$^{-1/3}$) characterizing the rough bottom properties.

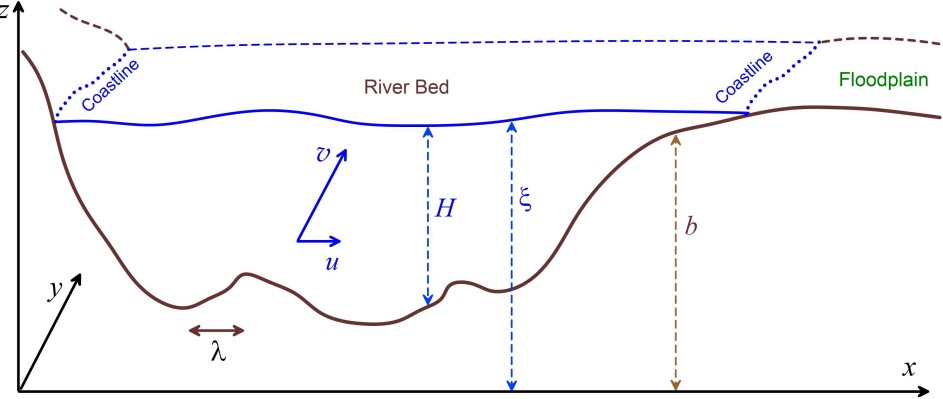

**Figure 7.** Geometric scheme of a single-layer model of shallow water. The main river channel is directed along the $y$-coordinate. The terrain height $b(x, y)$ depends on coordinates and does not depend on time. The water level is $\xi = b + H$ ($H$ is the depth of the liquid).

It should be noted that the quantity $n_M$ has a phenomenological character and the model (6) in the form:

$$n_M = n_0 + \sum_{k=1}^{K} n_k \tag{7}$$

is often used to describe a wide variety of physical factors that affect the hydraulic resistance of the flow in the channel in addition to the actual bottom roughness $n_0$, for example, meandering/branching of the channel ($n_1$), changes in the channel cross section ($n_2$), the presence of vegetation ($n_3$), drag due to sediment transport ($n_4$), inhomogeneity of the channel shape ($n_5$), presence of bottom ridges ($n_6$), unsteady flow ($n_7$) and other similar factors [4,6,39–42]. The diffusion approximation can be used to simulate the movement of water in the limiting case of strong resistance, for example due to dense aquatic vegetation [43].

We use conditions at the boundaries of the computational domain, which are described in detail in [44]. The most important requirement for a numerical algorithm for solving equations of fluid dynamics is the correct description of non-stationary wet–dry fronts inside the computational domain, including events of collisions of such fronts [37,45–48]. This is ensured by the use of well-balanced and positivity-preserving finite-volume methods for integrating the shallow water equations under conditions of significantly inhomogeneous topography [4,27,49]. This problem is solved using various approaches, for both single-layer models and two-layer (and multilayer) shallow water Equations [50]. We use the Combined Smooth Particle Hydrodynamics–Total Variation Diminishing (CSPH-TVD) method, which has the properties mentioned above [4,27,28,51].

The SDS-2DH method makes it possible to model horizontal large-scale vortices in shallow water [52]. A characteristic feature of the corresponding flows in the channel is

the formation of a vortices system that resist the flow [41,53]. Turbulent viscosity $\nu_{turb}$ is described by various approximations, for example, through the ratio of the squared kinetic energy of turbulent motions to the turbulent dissipation rate (see [54,55] and references therein) or by using the Smagorinsky approach [56]. For wide bodies of water, large eddies are able to redistribute both sediment and pollutants over long distances [51,57].

The influence of the wind is calculated through the wind stress $\tau_w \propto \varrho_a c_{10} W_{10}^2$ (where $\varrho_a$ is the air density, $c_{10}$ is the coefficient of wind drag, $c_{10} = 1.45 \cdot 10^{-3}$ and $W_{10} = W$ is the wind speed at a height of 10 m from the water surface). In general, the force of friction between air and water flow depends on the difference in the speeds of the two media [58]:

$$\mathbf{f}^{(wind)} = C_a \frac{\varrho_a}{\varrho H} \left(\mathbf{W} - \mathbf{u}\right) \cdot \left|\mathbf{W} - \mathbf{u}\right|, \tag{8}$$

where $\varrho$ is the liquid density and the parameter $C_a$ depends on the state of the water surface. Linear dependence on the velocity difference:

$$C_a = 10^{-3} \cdot \left(0.5 + 0.1 \cdot \left|\mathbf{W} - \mathbf{u}\right|\right), \tag{9}$$

is a satisfactory model for $C_a$ (velocities are measured in units of m·s$^{-1}$ m·s$^{-1}$).

### 2.3. Multilayer Numerical Model of the Floodplain Hydrological Regime

A characteristic feature of floodplain landscapes is the variability of the terrain, which increases significantly during flood events, introducing additional resistance to water flow due to sediment transport [28,59,60]. The second factor affecting surface water dynamics is groundwater [51,61–63]. The sediment movement is determined by the flow of the solid component $\mathbf{J}_b$, which leads to a change in the bottom profile with time $b(x, y, t)$ (Figure 8). Groundwater is located inside a porous layer ($\ell(x, y)$ is thickness), whose lower boundary is connected to a waterproof layer (such as clays). The transition of water from the surface layer ($H$) to groundwater aquifers is determined by the process of infiltration. Such two-layer and multilayer numerical models are very effective for various hydrological applications [28,63,64].

The self-consistent dynamics of surface water and sediment is described by the systems (2) and (3) together with the Exner equation:

$$(1 - \psi) \frac{\partial b}{\partial t} + \nabla \mathbf{J}_b = q_b, \tag{10}$$

where $\psi$ is the function of bottom porosity (relative fraction of air pores) and $q_b$ is the sediment function of sources ($q_b > 0$) and sinks ($q_b < 0$). The parameter $\psi$ in general case can depend on the coordinates, since the properties of the underlying surface vary widely in floodplain areas, $\psi = 0.1$–$0.5$.

The $\mathbf{J}_b$ flux is determined by the balance of complex erosion processes, soil transport and sedimentation [28,65]. The susceptibility of soil to erosion is characterized by the integral parameter of soil erodibility [66], which strongly depends on the aggregated soil structure, in particular, the distribution of particle size, chemical composition and organic matter content, as well as slope [67]. These same factors affect hydraulic conductivity. Erosion due to precipitation can make a significant contribution to the $q_b$ source in (10). The processes of soil erosion and infiltration are interconnected in the near-surface layer. The bottom layer in a water reservoir is sensitive to unsteady flow fluctuations, increasing the concentration of suspended particles and enhancing sediment transport [66–68].

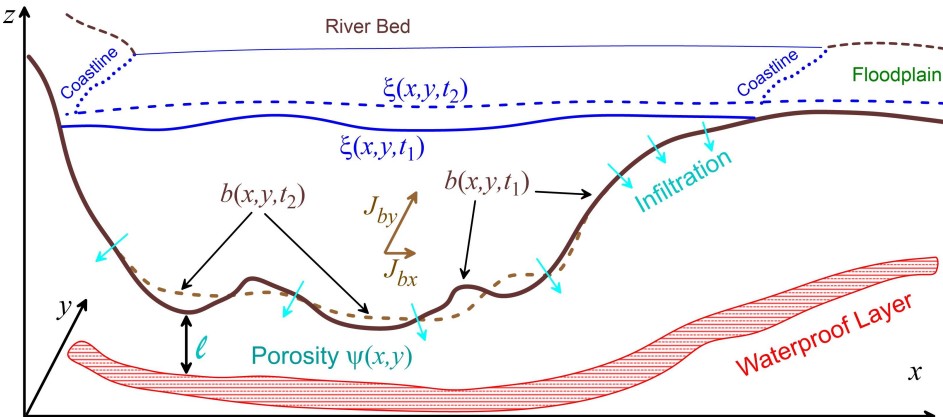

**Figure 8.** Geometric scheme of the three-layer model, including surface water, sediment and groundwater. The solid and dashed lines show the bottom profiles $b$ for two different times $t_1$ and $t_2$. The red color indicates the position of the waterproof layer, which is the groundwater bed.

Two-dimensional sediment models allow efficient modeling of sediment dynamics and morphological processes [11,69–71]. We use the Darcy model for groundwater (see [62] and references there), which leads to the Richards equation. This equation has a diffusion type and allows modeling both saturated and unsaturated porous media [72].

The movement of water in an aquiferous porous underground layer is described by the Richards Equation: [63,72,73]

$$\frac{\partial \psi \varrho_g}{\partial t} = \nabla \left[ K(\nabla p - \mathbf{g}\varrho_g) \right],$$

(11)

where $K$ is the hydraulic conductivity, for example, $K = k_s \cdot (\Theta/\Theta_{\max})^{\alpha_s}$ (where $\Theta$ is the ratio of liquid volume to soil volume and $\Theta_{\max}$ is the soil porosity), nonlinear saturation is described by the soil pore-disconnectedness index $\alpha_s < 1$ [73], $p$ is the pressure and $\varrho_g$ is the volume density.

We use the floodplain hydrological modeling software developed by [26–28,44,51] using up-to-date spatial data, characterizing the heights of the relief and the physical properties of the underlying surface. Figure 9 shows the general picture of flooding of the northern part of the Volga-Akhtuba floodplain for various hydrographs at the time of its peak. The hydrological regime of spring flooding is characterized by the following [2,74]:

1. Water level rise in the Volga River by 6–10 m due to $Q^{(m)}$ growth by 4–5 times.
2. An increase in the rate of water discharge into the Akhtuba River $Q_A(t)$, which repeats the typical form of the dependence $Q^{(m)}(t)$ (see Figure 2) .
3. Outflow of water from the Akhtuba River through large main canals with subsequent filling of medium and further small channels, which ensures flooding of the interfluve plain. This way of water movement gives about 70% of the volume and area of flooding.
4. The direct outlet of water to the floodplain from the left bank of the Volga River through local relief depressions and small channels is responsible for about 30% of flooding.

Hydrological connectivity of the interfluve territory is of particular importance in the conditions of low peaks of spring hydrographs. Our calculations show that the critical value is the level $Q_1^{(m)} crit \simeq 16,000 \, \text{m}^3 \cdot \text{s}^{-1}$, at which the hydrological connectivity of the interfluve disappears on the main area. This estimate is consistent with observational data in 1996, 2006 and 2015, when the maximum water discharge was within $Q_1^{(m)}(1996) = 20,000 \, \text{m}^3 \cdot \text{s}^{-1}$ ($\theta_1 \simeq 2 \, \text{days}$), $Q_1^{(m)}(2006) = 18,000 \, \text{m}^3 \cdot \text{s}^{-1}$ ($\theta_1 \simeq 3 \, \text{days}$), $Q_1^{(m)}(2015) = 16,000 \, \text{m}^3 \cdot \text{s}^{-1}$ ($\theta_1 \simeq 6 \, \text{days}$), which led to ecological disasters in the floodplain during these seasons.

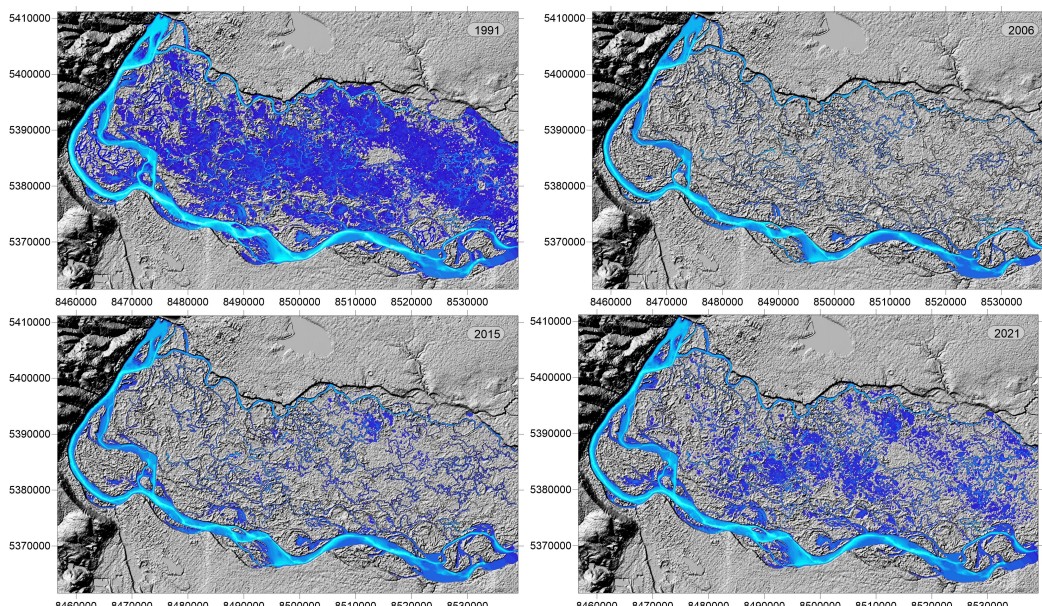

**Figure 9.** Examples of the VAF flooding dynamics calculations for hydrographs of different years.

The technological basis of our further research is the ECOGIS software for modeling the hydrological regime of a real territory with complex topography [4,26–28,30,51]. The basic module for surface water modeling is described in [4,27]. Papers [28,51] are devoted to self-consistent accounting for sediment and groundwater dynamics. Features of high-performance computing parallelization are discussed in [26,28]. Modules for working with digital elevation models and spatial data were developed in [29,30] in the application for hydrological calculations.

### 3. Modeling the Floodplain Territorial Structure: Methods, Instruments and Technology Research

We study the dynamics of the complex structure of the territory of the floodplain ESES, which is a superposition of its hydrological ($\mathcal{H}$) and functional ($\mathcal{F}$) structures. The elements of the hydrological ($\mathcal{H}$) structure are sites of the territory with the same $\mathcal{H}$-type, determined by the frequency range of their floodings. The number and size of flood frequency ranges are determined by the objectives of the study. We use aggregated enlarged ranges characterizing: (1) unflooded, (2) unstable flooded and (3) stable flooded lands, which we call $\mathcal{H}$-types.

The stable $\mathcal{H}$-structure of the floodplain territory is the basis for constructing its effective $\mathcal{F}$-structure in the process of socio-economic development and the ESES creation. The subsequent natural or natural-technogenic change in the $\mathcal{H}$-structure causes a decrease in the efficiency of the $\mathcal{F}$-structure of the floodplain area. The presence of a forecast of long-term changes in the $\mathcal{H}$-structure makes it possible to design strategies for the socio-economic development of the floodplain ESES, whose purpose is to restore and maintain a balance between the $\mathcal{H}$-structure and $\mathcal{F}$-structure of the territory through the coordination of hydrotechnical projects and socio-economic development projects [7,9,74,75].

The stability of the $\mathcal{H}$-structure of a territory is determined by the constancy of its main factors: the relief, the channel system and the distribution function of the water discharge volume during the spring flood. The variability of these factors may be the reason for the change in the $\mathcal{H}$-structure. The construction and further operation of the VHPS caused a change in all three factors of the $\mathcal{H}$-structure of the VAF. Observational data show a relative decrease in the average volume of controlled spring floods by 20% or more compared to natural levels before 1960. This is the reason for the progressive decrease in the depth of the Volga River below the dam, as well as the average depth of small channels in the interfluve. The socio-economic development of the VAF territory has significantly changed

its relief: more than 200 lakes have been destroyed, more than 300 km of highways have been built, several dozen new settlements have appeared and the agricultural area has increased significantly in the last 60 years.

Most of these changes took place in the 1960s. The created $\mathcal{F}$-structure of the VAF territory best corresponded to its $\mathcal{H}$-structure in that period. Therefore, the main object of our study is the change in the $\mathcal{H}$-structure of the VAF in subsequent decades. During this period, there were no significant changes in the $\mathcal{F}$-structure of the territory of the VAF, and, consequently, in the relief of its territory.

The study of the influence of changes in the depth of small channels on flood dynamics in the VAF showed that it is significant only for floods with small volumes occurring with a frequency of less than 20% [76]. Therefore, the degradation of small channels does not affect the studied frequency ranges, which characterize our three types: unflooded, unstable flooded and stable flooded territories. Thus, we study the change in the $\mathcal{H}$-structure of the VAF over the period 1962–2020 and build forecast up to 2050, assuming that the main factor is the slow change in the geometric characteristics of the Volga riverbed downstream of the dam, which is happening now and will continue to occur in the future.

The sample size $T$ (the number of consecutive years of observations) determines the type and parameters of the flood frequency distribution function ($\phi(V)$) and should be large enough to provide a given error, which decreases with increasing $T$. On the other hand, this size should make it possible to demonstrate the stability of this distribution function and reveal the dynamics of the $\mathcal{H}$ structure of the VAF over $N$ years (the entire observation period) due to changes in the geometric parameters of the Volga riverbed. We determine the value of $T$ experimentally based on statistical processing of the real observations and the results of hydrodynamic simulations of floods in the interfluve area for 1962–2021.

The algorithm for constructing a $\mathcal{H}$-structure and estimating its accuracy is the following. First, the volumes of spring discharge at the first stage ($V$ for the $\tau$-th year) and the corresponding $N$ digital maps of the areas of maximum annual flooding of the floodplain are determined based on the results of modeling flood water dynamics in the northern part of the VAF with $N$ real hydrographers. Preliminary values of the frequency ranges of flooding are determined by experts based on the purpose of the simulation. The sample size $T$ is determined by analyzing the scatter of two coefficients of lognormal distributions $(N, T)$ approximating $N - T$ frequency distributions for $T$ volumes of flood hydrographs in the form:

$$\phi(V) = \frac{1}{\sqrt{2\pi}\,\sigma(r, T)\,V} \cdot \exp\left\{ -\frac{[\ln(V) - \mu(r, T)]^2}{2\,[\sigma(r, T)]^2} \right\}, \tag{12}$$

which are characterized by two parameters: $\mu$ and $\sigma$ ($r = 1, \ldots, N - T$). Each pair of these parameters characterizes a sample of consecutive elements of the general population of $N$ elements. The first element has the number $r$, and the last one is $r + T$ ($T = 1, \ldots, N$). The number of samples is $N - T$ for a fixed value of $T$. The lognormal distribution (12) is traditionally used in hydrology to model the volume distribution of river flood discharges, which depends on two multiplicative factors, for example, the amount of snow falling in the river basin in winter and the intensity of the snowmelt regime (almost all water enters the river at high intensity and a significant amount of it is absorbed into the soil at low intensity) [77,78].

Each of the $N - T$ versions of the $\mathcal{H}$-structure of the floodplain area (digital maps) is built on the basis of the corresponding $T$ digital maps of the maximum annual flood corresponding to each of the $N - T$ sets of flood hydrographs for $T$ consecutive years from the total observation period of $N$ years. We define pairs of boundary elements as the union and intersection of the corresponding $N - T$ fragments of digital maps for each $\mathcal{H}$-element. The areas ratio of these boundary elements serves as a measure of the error in determining the corresponding $\mathcal{H}$-element. The required accuracy is achieved by increasing $T$ and/or reducing the number of elements of the $\mathcal{H}$-structure, if necessary.

The $\mathcal{F}$-structuring of the floodplain territory is based on its ESE-cadastral map, which divides the territory into sections with a fixed kind of use. The elements of the $\mathcal{F}$-structure of a territory are its fragments of one kind of use ($\mathcal{F}$-type), corresponding to one or more cadastral parcels. Each kind of use is assigned to one of three types: social, economic or ecological. The elements of the complex structure of the floodplain territory are functionally and hydrologically homogeneous fragments of its territory, each of which belongs to some $\mathcal{F}$-kind and $\mathcal{H}$-kind. As elements of the aggregated complex structure, we consider functionally and hydrologically homogeneous fragments of its territory, each of which belongs to some $\mathcal{F}$-type and $\mathcal{H}$-type, so that the aggregated complex structure consists of only 12 elements.

The second tool of our study is the regression model of the slow dynamics of the parameters of the flood hydrological regime in the interfluve. These changes are caused by slow progressive erosion of the bottom of the main river channel of the Volga below the VHPS, which leads to a decrease in the level of flood waters and, accordingly, a reduction in the area of maximum annual flooding. Construction of the dependence regression of the flood water level $\xi$ on the flood water discharge $Q(\tau)$ (HPS hydrograph) $\xi(Q_1^{(m)}(\tau),\tau)$ ($\tau$ indicates the year and $\tau = 1961 + r$ for VAF) on the basis of long-term observations allows us to construct a reconstruction of the $\mathcal{H}$-structure of both past decades and its forecast for future decades.

We use algorithm for constructing a predictive $\mathcal{H}$-structure and estimating its accuracy under two assumptions: 1) the constant climatic conditions that determine the future distributions of spring flood volumes and 2) the possibility of describing the riverbed morphology dynamics by linear regression of the flood water level below the dam.

The lognormal distribution (12) is used to approximate each frequency distribution of spring flood volume corresponding to one of the $r = N - T$ sets. The result is four limiting lognormal distributions with parameters:

$$
\begin{aligned}
&(\mu_i, \sigma_j)\ (i,j = 1,2)\,; \\
&\mu_1 = \max_r \mu(r)\,,\ \mu_2 = \min_r \mu(r)\,; \\
&\sigma_1 = \max_r \sigma(r)\,,\ \sigma_2 = \min_r \sigma(r)\,.
\end{aligned}
\tag{13}
$$

We generate $k = 5$ corresponding sets of $T$ flood hydrograph volume values.

The transition from estimated forecast $4Tk$ values of the volumes of flood hydrographs to estimated forecast values of the parameters of their first stages occurs on the basis of a linear regression constructed for the array of values $Q_1^{(m)}(\tau), \theta_1(\tau)\ (\tau = 1, \dots, N)$ for the observation period. For such $4Tk$ estimated forecast flood hydrographs, $4Tk$ estimated digital maps of maximum flooding of the floodplain area and $4k$ estimated digital maps of its $\mathcal{H}$-structure are constructed. The forecast error of each element of the $\mathcal{H}$-structure is defined as the ratio of the areas of its upper and lower boundaries—the union and intersection of the fragments of the $4k$ evaluation digital maps corresponding to it. The largest of the errors is considered the total forecast error. If the error of some elements exceeds the allowable level, then the frequency ranges corresponding to it are enlarged, and a new $\mathcal{H}$-structure is constructed and the accuracy of its forecast is studied.

The third tool of our analysis is the model of criteria for the state of the floodplain territory. We, as experts, specify a family of characteristic functions on the set of types of elements of the $\mathcal{F}$-structure, which allows us to calculate the degree of harmonicity of each $\mathcal{H}$-type to each $\mathcal{F}$-type. Expert analysis of the hydrological conditions for the effective functioning of the main types of elements of the floodplain $\mathcal{F}$-structure made it possible to construct a family of characteristic functions presented in Table 1:

$$
f_1(n) = \begin{cases}
0, & if \quad 0 \le n < n_1\,(1-\varepsilon); \\
\dfrac{n}{\varepsilon} - \dfrac{1-\varepsilon}{\varepsilon}\,n_1, & if \quad n_1(1-\varepsilon) \le n \le n_1; \\
n, & if \quad n_1 < n \le 1,
\end{cases}
\tag{14}
$$

$$f_2(n) = \frac{n+1}{2}, \tag{15}$$

$$f_3(n) = \begin{cases} 1 - \dfrac{2n}{\varepsilon}, & if \quad \varepsilon \geq 2n; \\ 0, & if \quad \varepsilon < 2n, \end{cases} \tag{16}$$

$$f_4(n) = \begin{cases} 1 - \dfrac{n}{\varepsilon}, & if \quad \varepsilon \geq n; \\ 0, & if \quad \varepsilon < n, \end{cases} \tag{17}$$

where $n$ is the flood frequency and $n_1$ is the expert assessment of the threshold frequency of flooding, describing the stable existence of the floodplain ecosystem and forests. The functions $f_1$ and $f_2$ decrease with decreasing frequency of flooding and characterize the possibility of existence of various kinds of zones of ecological type by stable flooding. Increasing functions $f_3$ and $f_4$ with decreasing frequency of flooding characterize the level of damage to different kinds of zones of social and economic types from the flood. The $\varepsilon$ parameter is a tuning tool based on financial and economic evaluation. The expert threshold value of the flooding frequency $n_1$ determines the stable existence of the floodplain ecosystem.

**Table 1.** Kinds and types of the main elements of the $\mathcal{F}$-structure of the floodplain territory and characteristic functions of the criteria of state.

| | Kind and Type of Element of $\mathcal{F}$-Structure | Social Criterion | Environmental Criterion | Economic Criterion |
|---|---|---|---|---|
| 1 | Wetlands (ecological type) | $f_1(n)$ | $f_1(n)$ | $f_1(n)$ |
| 2 | Water meadows (economic type) | $f_1(n)$ | $f_1(n)$ | $f_1(n)$ |
| 3 | Forests (ecological type) | $f_2(n)$ | $f_2(n)$ | $f_2(n)$ |
| 4 | Recreational areas (social type) | $f_2(n)$ | 0 | 0 |
| 5 | Urbanized zones (social type) | $f_3(n)$ | 0 | 0 |
| 6 | Economic zones, including zones of irrigated agriculture (economic type) | 0 | 0 | $f_4(n)$ |

Thus, a simple three-element $\mathcal{H}$-structure of a floodplain area has the following form: unflooded areas ($n = 0$), unstable flooded areas ($0 < n < n_1$), stable flooded territories ($n_1 \leq n \leq 1$). The aggregated complex $\mathcal{HF}$-structure of the floodplain area can be represented as a set of aggregated elements of nine types as a result of the combination of kinds of elements of the $\mathcal{F}$-structure into social, ecological and economic groups (types). The values of social, ecological and economic criteria of the state of each element of the complex territorial structure are calculated as the product of the corresponding area and the values of the corresponding characteristic functions. The aggregate values of the criteria are obtained by summing their values over all structural elements of each type.

The change in the frequency of flooding of a significant number of elements as a result of riverbed dynamics is the reason for the change in their characteristic functions and, consequently, the criteria for their state and the state of the entire ESES of the floodplain area.

## 4. Results of Modeling the Dynamics of the ESES Territorial Structure for the Northern Part of the Volga-Akhtuba Floodplain

The presence of a territory with an indefinite type of use ($\mathcal{N}$-type) is a feature of the $\mathcal{F}$-structure of the VAF. Figure 10 shows the modern aggregated territorial $\mathcal{FN}$-structure of VAP for four types.

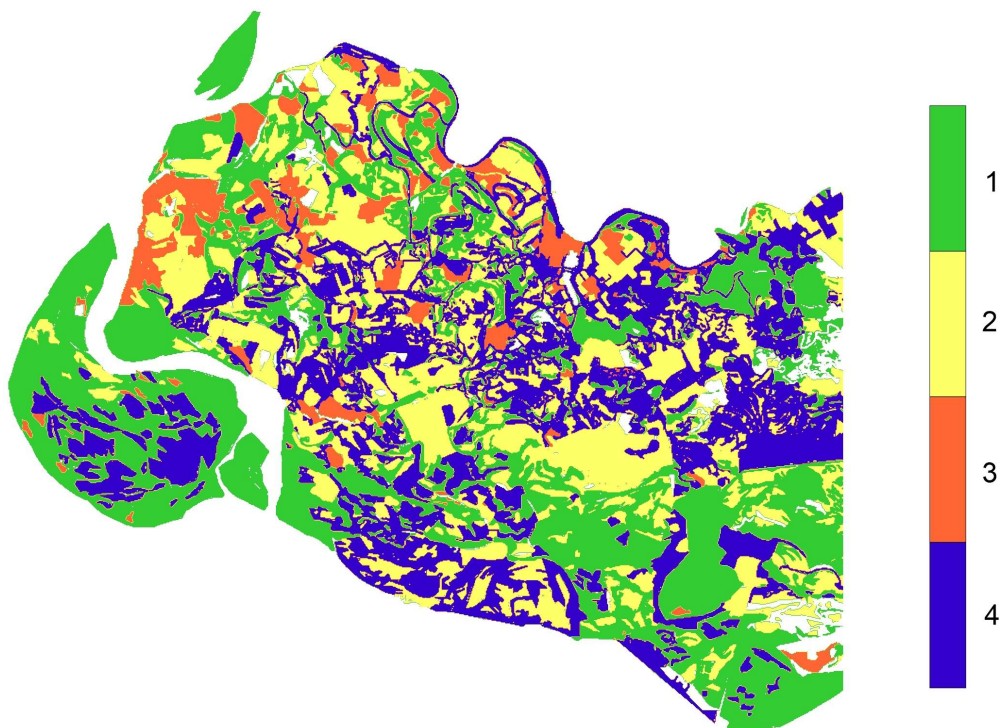

**Figure 10.** The modern $\mathcal{FN}$-structure of the VAF: 1—ecological territories, 2—economic territories, 3—social territories, 4—territories with an uncertain cadastral type.

We have built maps of the maximum flooding of the northern part of the VAF for each VHPS flood hydrograph in the period 1962–2021 based on a series of computational experiments. The virtual values of the parameters of the first stage of the flood (water discharge $Q_1^{(m)}$ and duration $\theta_1$, See Figure 2) corresponding to these maps for the period 1962–2002 are fitted, as well as the family of lognormal distributions of these parameters for various sample sizes $T$ is constructed (Figure 11).

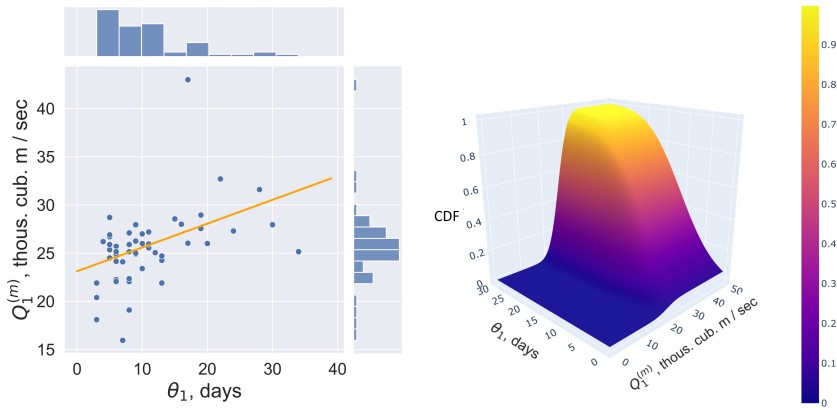

**Figure 11.** Virtual values, density and distribution function of the first stage parameters of the flood hydrograph through the VHPS for 1962–2021. The orange line shows the linear regression of $Q_1^{(m)}(\theta_1)$.

Figure 12 shows the results of constructing the distribution functions of flood water volumes through VHPS for 1962–2021 using the methodology in Section 3. Dependences of the values and confidence intervals of the parameters of the first stage of the flood hydrograph on $T$ average values (over $T$ years) are shown in panel (a) for 1962–2021. Panel (b) in this figure shows three examples of flood volume frequency distributions for $T = 30$. The dependences of the values and confidence intervals of two parameters of the lognormal

distribution of the normalized volume of the first stage of the flood hydrograph on $T$ (for $T$ years) are shown in panel (c) for 1962–2021.

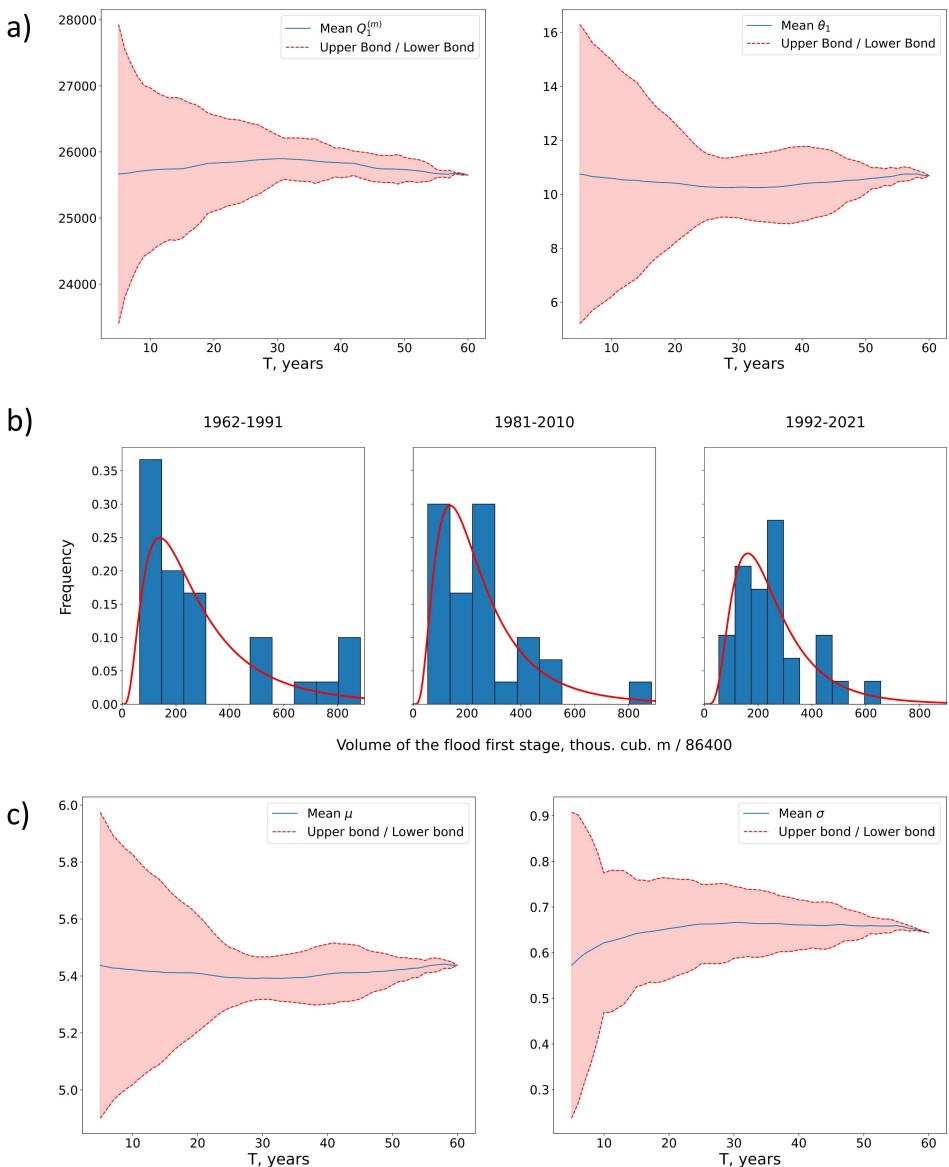

**Figure 12.** (**a**) Dependences of the values and confidence intervals of the hydrograph parameters for the Volgograd HPP at the first stage on $T$ for 1962–2021; (**b**) coefficients of lognormal distributions of the normalized volume of flood waters; (**c**) examples of frequency distributions for $T = 30$.

The construction of the $\mathcal{H}$-structure is based on series of computational experiments for $\tau \in [1962; 2030]$, $T \in [20 : 40]$. We use the values $n_1 \in [0.75 : 0.85]$, $\varepsilon = 0.1$. The value $n_1 = 0.85$ is assumed by default below. Figure 13 shows the relative areas of the sum of the elements of an aggregated 12-element modern complex $\mathcal{HFN}$-structure for the VAF territory ($\tau = 2000$, $T = 20$, $n_1 = 0.85$). The error in its determination (the maximum of the errors in the determination of its elements) is 4%.

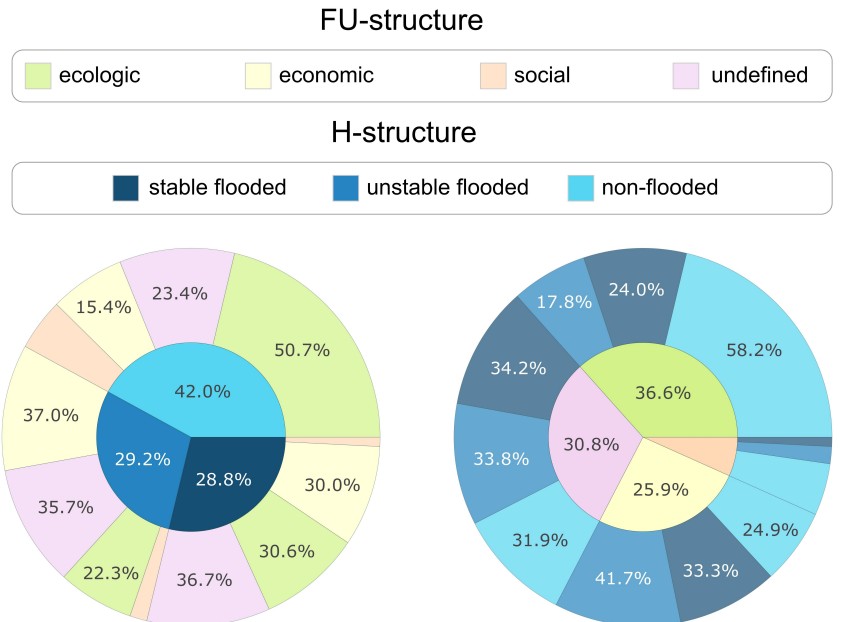

**Figure 13.** Relative total areas of 12 types of modern territorial $\mathcal{HFN}$-structure for the northern part of the VAF.

Figure 14 shows digital maps of the modern VAF $\mathcal{HFN}$-structure. This complex structure defines 12 different complex elements (territory fragments) highlighted in different colors and shades, each of which belongs to one of three types of $\mathcal{H}$ structure and one of four types of $\mathcal{FN}$ structures.

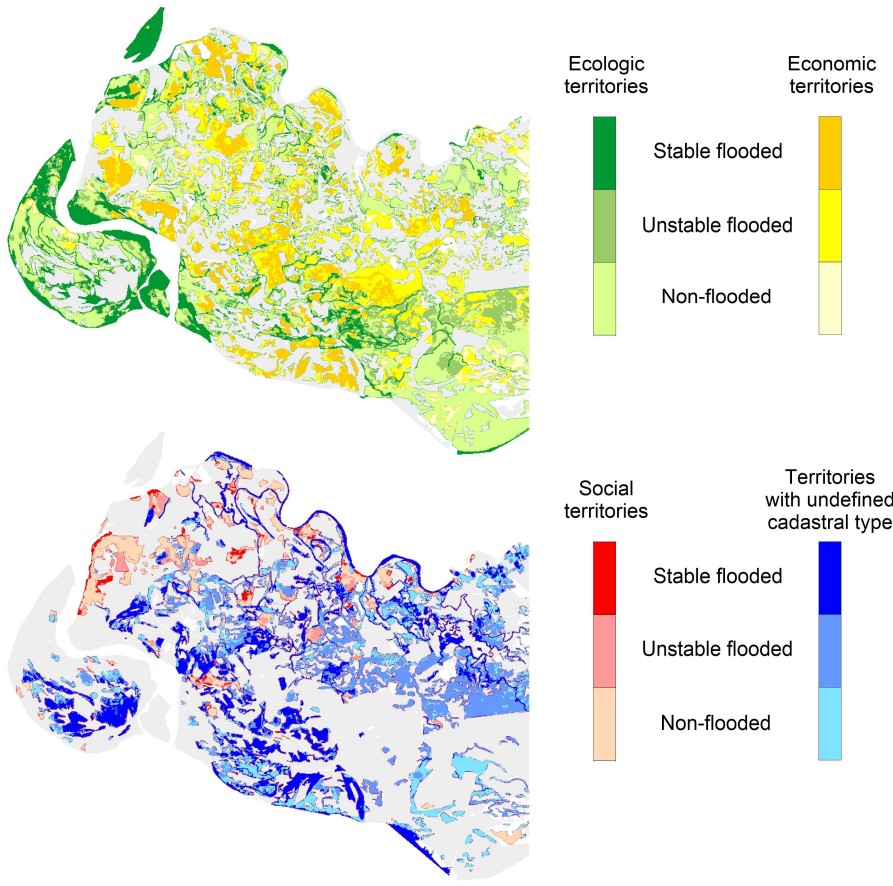

**Figure 14.** Maps of the complex structure for the northern part of the VAF in 2021.

The slow erosion of the Volga river bed below the dam is the reason for the progressive decrease in the level of flood waters, which has now reached 1.2–1.3 m compared to 1962. We used the time series of observations of flood water heights $\xi(Q_1^{(m)}(\tau), \tau)$ ($\tau = 1962, \ldots, 2021$) at the gauging stations to build a virtual hydrograph that takes into account the lowering of the flood water level for modeling of retrospective and forecast flood maps of past and future decades on the modern channel structure:

$$Q_{virt}(\tau, \tilde{h}) = 6020.8 + 1795.3\,\tilde{h} + 176.9\,\tilde{h}^2\,, \tag{18}$$

where $\tilde{h} = \xi - \xi^{lw} + \Delta h(\tau)$, $\xi$ is the water surface level of the source of the Akhtuba river according to the data of gauging stations and $\xi^{lw}$ is the low water value, $[Q_{virt}] = \mathrm{m}^3 \cdot \mathrm{s}^{-1}$, while the linearly decreasing function for calculating corrections is as follows:

$$\Delta h(\tau) = 41.55 - 0.021 \cdot \tau \quad [\mathrm{m}]\,, \tag{19}$$

We use the formulas (18) and (19) to construct a retrospective (for the interval 1962–1991) and forecast (for the interval 2021–2050) $\mathcal{H}$-structures of the territory.

Figures 15 and 16 show the dynamics of the areas of the VAF stably flooded area and its ESES state criteria for the interval 1991–2050. The ecological criterion demonstrates a decreasing trend, the economic and social criteria slightly change relative values normalized to the corresponding values for 1991, while the relative area of the steadily flooded territory as a whole shows a significant drop. A characteristic feature of the situation is a noticeable increase in the rate of decline after 2008, which, apparently, is due to sharp climatic changes in the Russian Plain (Volga River drainage basin) after 1978, taking into account our sample size of $T = 30$ years.

We studied the variability of the parameters of the lognormal distribution of the values of the relative volume of the flood hydrograph of the VHPS for different 30-year ranges over the observation period 1962–2021 in order to construct a forecast of $\mathcal{H}$-structure for the interval 2022–2050. Figure 17a shows the parameters of the marginal lognormal distributions, their average value over the period 1962–2021 and the spread in the range of one standard deviation.

We generated 2 ($k = 2$) random sets of 30 flood hydrograph volumes $V$ for each of the marginal distributions. The transition from $V = Q_1^{(m)}\theta_1$ to the forecast for the estimated values of the parameters of flood hydrographs $(Q_1^{(m)}, \theta_1)$ was carried out using linear regression $Q_1^{(m)} = 237 \cdot \theta_1 + 23117$, which is built on the basis of hydrograph data for 1962–2021 (see Figure 11). The result of the forecast is $4Tk = 240$ estimated pairs of parameters of the flood hydrographs of the hydro station $(Q_1^{(m)}, \theta_1)$, taking into account the replacement $Q_1^{(m)}(\tau)$ by $Q_{virt}(\tau)$ ($\tau = 2021, \ldots, 2050$), which made it possible to construct 240 estimated digital maps of the maximum flooding of the floodplain area and 8 estimated digital maps of its $\mathcal{H}$-structure. These data provided the calculations of the upper and lower bounds of the forecast, which set the forecast error for the ecological, economic and social criteria for the state of the floodplain territory and also the area of stable flooding for the entire forecast period. Figure 15 shows the results of these calculations. Figure 18 shows the area that has lost its stable flooding regime. The peculiarities of the behavior of forecasts in Figures 15 and 16 after 2045 are due to linear extrapolation (19), which leads to an increase in the sensitivity of the flood area to natural hydrograph fluctuations. Therefore, the forecast interval should be limited to 20–25 years. Figure 19 shows our forecast of the stable flooding area for the northern part of the VAF. "Intersection stable area" is a stable flooding area on all $4k = 8$ maps, "Union stable area" is a stable flooding area on at least one of the eight maps.

The use of the "Union stable area" forecast makes it possible to obtain a guaranteed estimate of changes in the complex structure of the VAF. Figure 20 shows a diagram of the relative areas dynamics of the elements of the complex structure for the northern interfluve. Figure 21 shows part of the stable flooded functional elements of the territory in the modern

complex structure, which was included in the corresponding unstable flooded functional elements in the complex structure forecast for 2050.

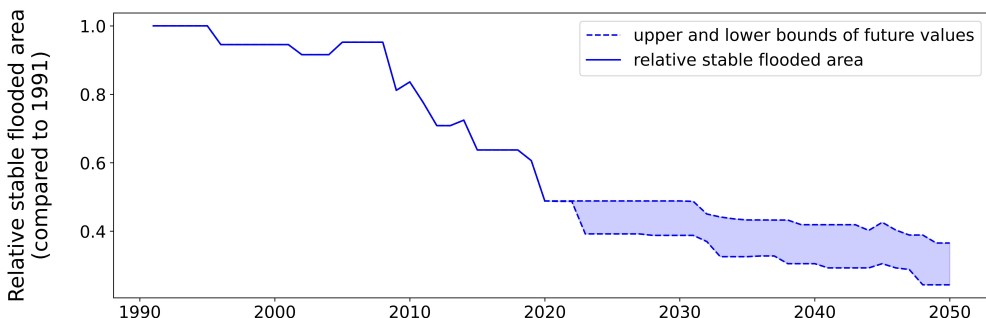

**Figure 15.** Retrospective (1991–2021) and forecast (2022–2050) dynamics of the area of stable flooded territory in the interfluve.

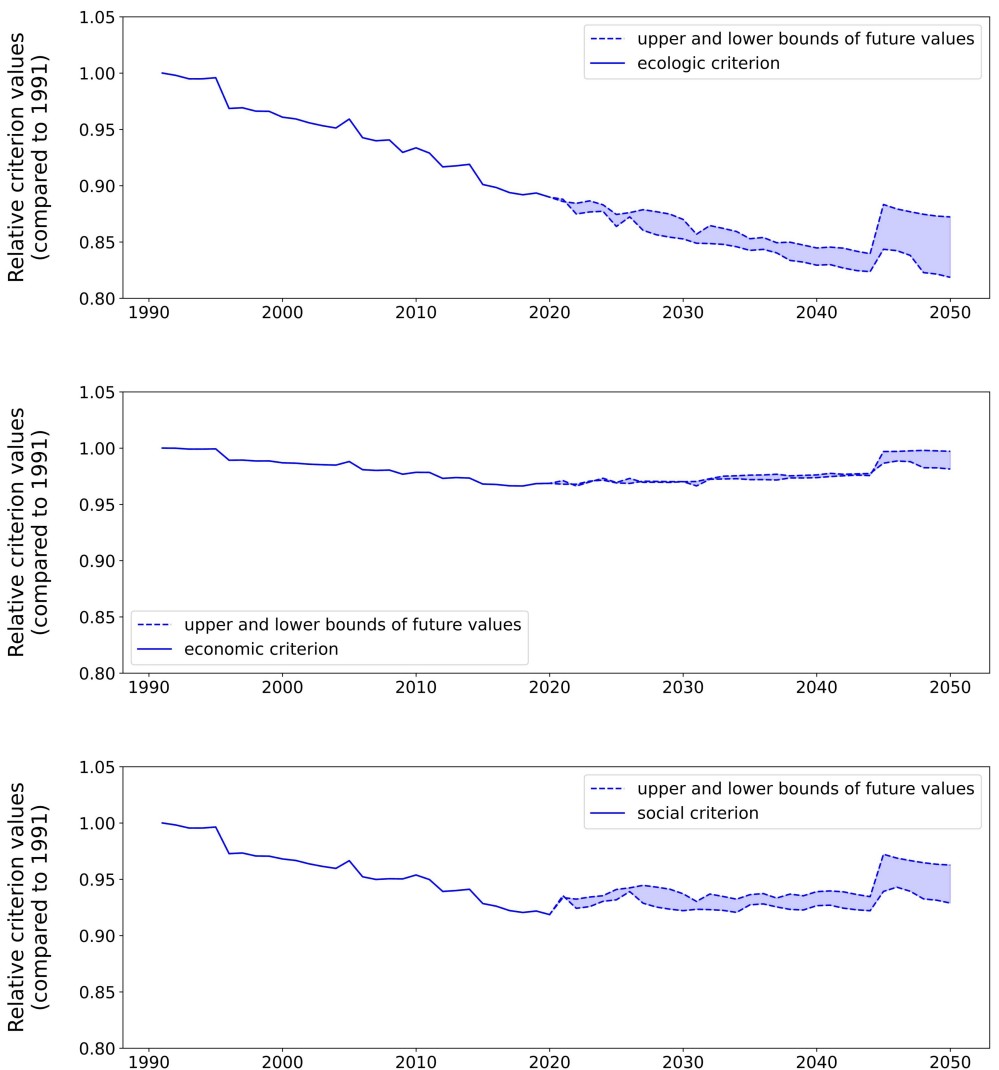

**Figure 16.** Retrospective (1991–2021) and forecast (2022–2050) dynamics of ESES at $\varepsilon = 0.1$ and $n_1 = 0.85$.

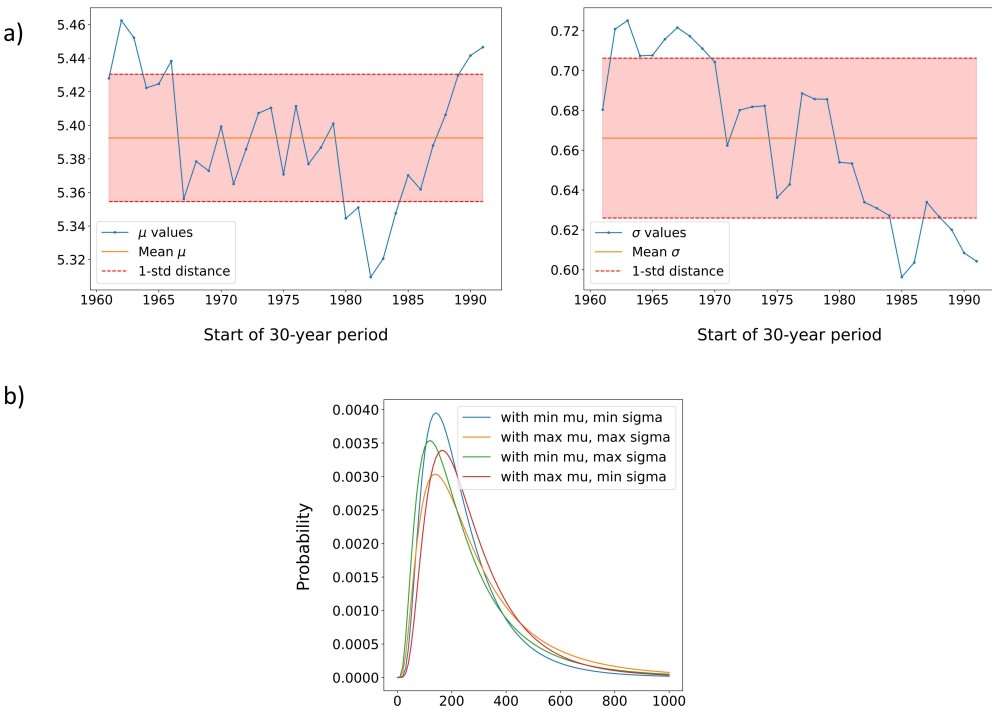

**Figure 17.** (**a**) Variability of the parameters of the lognormal distribution of the values of the relative volume of the flood hydrograph through the dam for different 30-year intervals. The color indicates the range of one standard deviation. (**b**) Limiting lognormal distributions of the actual volumes of flood hydrographs for 1962–2021.

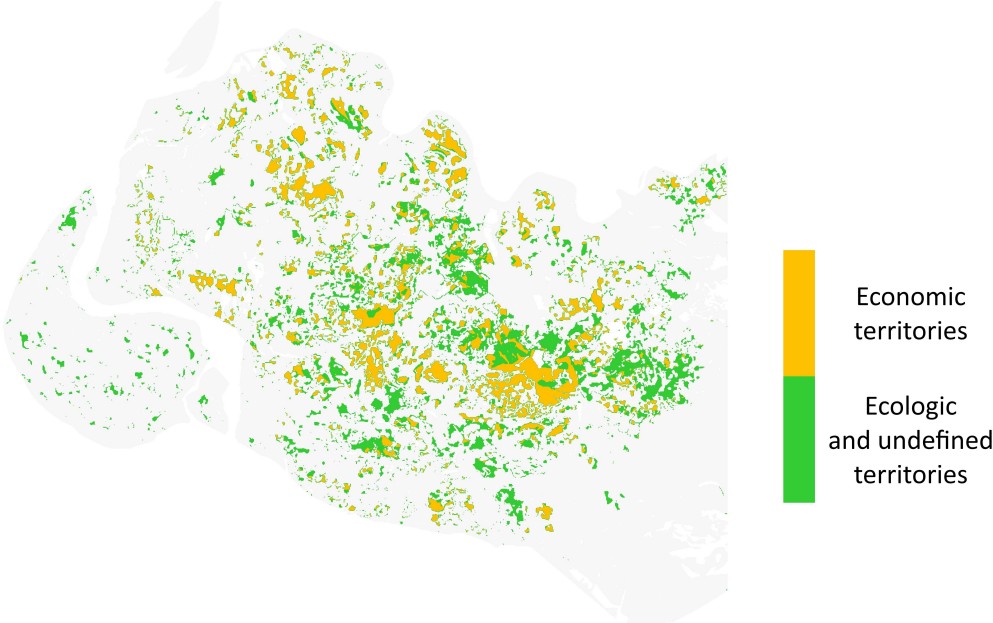

**Figure 18.** The territory that lost the stability of flooding for the period 1991–2021.

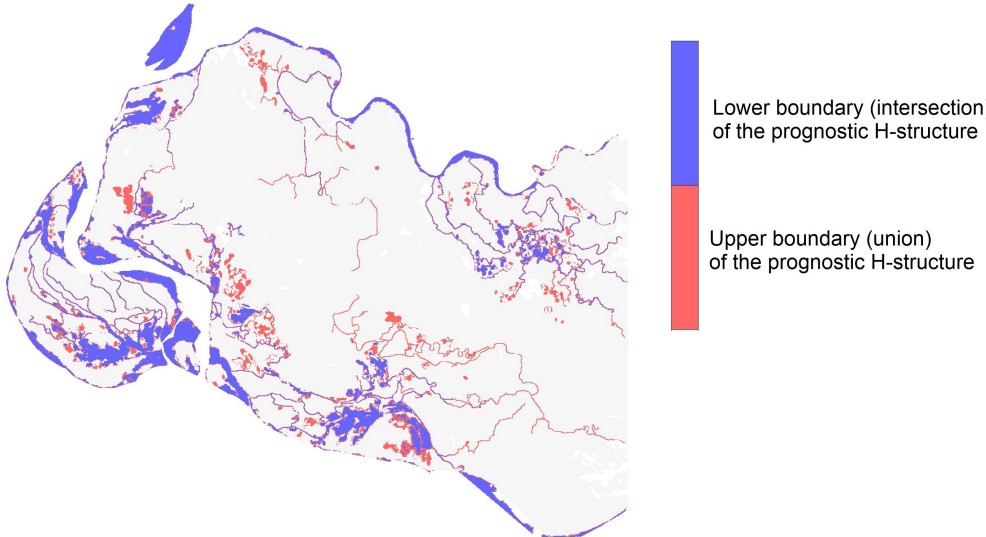

**Figure 19.** Forecast map of stable flooding of the VAF. The blue color shows the Intersection stable area, while the red color highlights the difference between the intersection stable area and the union stable area.

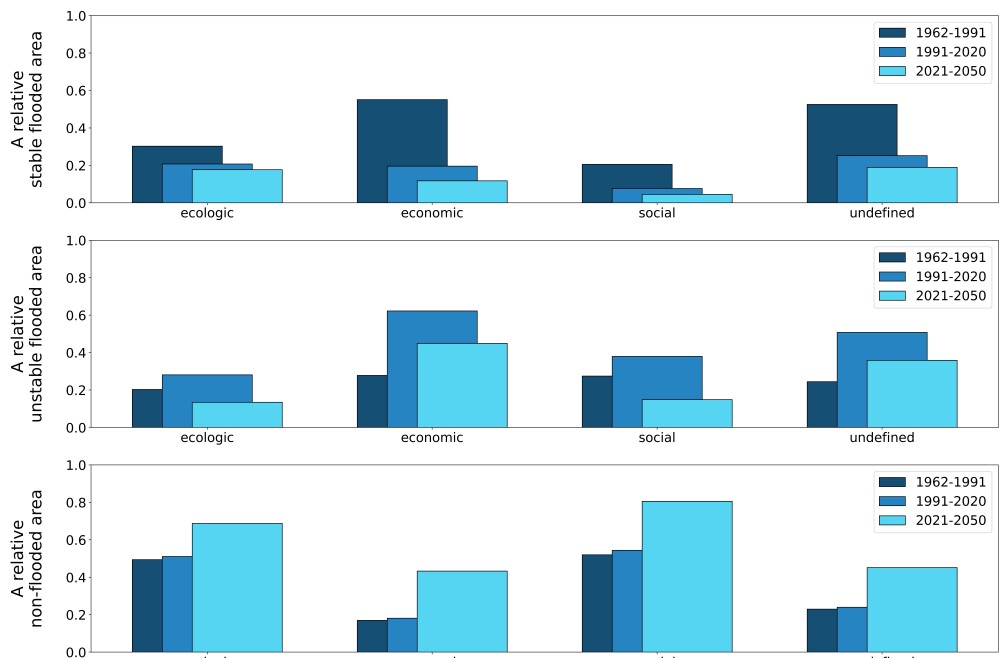

**Figure 20.** Diagrams of the elements dynamics of the complex structure for $T = 30$.

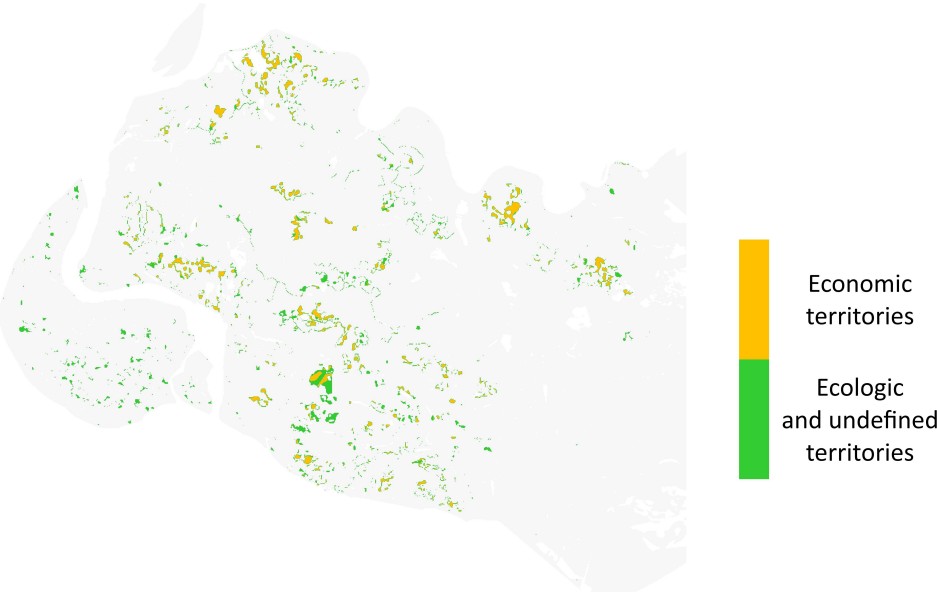

**Figure 21.** The area that will lose its flooding stability in the period 2022–2050 according to our forecast.

## 5. Discussion

Restoring the floodplain landscape to its natural state cannot be a realistic task due to the active use of the agricultural, recreational, energy and urban potentials of these territories in the face of increasing global climate change. Therefore, solutions must be found for the balance of opposing interests of different actors and the sustainability of the biological diversity of natural communities [79]. Maintaining such a balance requires forecasting the development of the situation in the conditions of natural and technogenic degradation of the floodplain area. Reliable forecasts are also needed to evaluate the long-term effectiveness of environmental and social projects.

The problem of structuring floodplain territories has been the subject of numerous articles in recent years. The traditional tasks of structuring are the determination of natural and functional differences [80–84], the effects of emergency floods [85–88] and the degree of anthropogenic degradation of ecosystems [84]. The most widely used are field methods [82,85,87], the results of processing data from satellite observations [83,84], methods of hydrodynamics [85,86] and geoinformation modeling [80,81,83–85,87,88], statistical analysis [83–88] and forecasting [86,88]. The novelty of our approach lies in the construction of a multidimensional structure that combines the hydrodynamic and socio-natural economic properties of territorial elements. Therefore, we use all the above grounds for structuring and research methods. Complementing these methods with expert characteristic functions for evaluating structural elements makes it possible to assess the state of the floodplain environmental and socio-economic system.

The main novelty of our approach lies in the method of modeling the dynamics of the territorial structure. Our method has three parts. The first part is an analysis of the minimum amount of observational data necessary both to achieve an acceptable accuracy in constructing the flood volume distribution function and to ensure the stability of the found territorial structure in the absence of an external dynamics factor. The second part is based on the model of the external factor of the complex structure dynamics, allowing its reconstruction several decades ago and its forecast several decades ahead. The third part includes a method for assessing the accuracy of forecasting the dynamics of the territorial structure, providing the calculation of upper and lower estimates of the boundaries of changes in its elements.

The proposed approach to modeling the dynamics of the complex structure of flooded territory can, in our opinion, serve as a universal basis for solving a large number of problems of floodplain areas.

The correctness of the forecast of the $\mathcal{H}$-structure change under the influence of negative factors is based on its stability in the absence of these factors. Therefore, the analysis of the stability of the $\mathcal{H}$-structure of the VAF is a large part of our study. We evaluate the error of our calculation algorithms and the natural variability of natural processes separately. The construction of $\mathcal{H}$-structures of the VAF with different frequency ranges and parameters $T$ and $\tau$ indicates that the natural variability of the boundaries of the elements of the territorial structure is determined by the variability of river hydrograph parameters, terrain features and frequency ranges. The accuracy of determining each structural element increases with an increase in its area and the width of the frequency range corresponding to it. The accuracy of determining all elements in all structures increases significantly with $T$ from 10 to 20, and insignificantly increases with $T$ from 20 to 40. The boundaries of a stably flooded area (with a frequency of more than 0.75) are determined most accurately; however, it decreases with an increase in $\tau$. The analysis of these results made it possible to establish that on the map of the retrospective $\mathcal{H}$-structure (1962–1991), relatively high unflooded areas are separated from stably flooded plains by narrow strips of slopes of an unstable flooded earth's surface. Therefore, a strong change in the height of flooding leads to a slight change in the boundaries of the structural elements of the VAF. On the maps of the modern (1992–2021) and forecast (2021–2050) $\mathcal{H}$-structures, a significant part of the boundary between stable and unstable flooded elements passes through flat areas. Therefore, even small changes in the height of the flood lead to significant changes in the indicated boundary.

We emphasize that the three-element $\mathcal{H}$-structure of the VAF corresponds best to its $\mathcal{F}$-structure. The stable flooded area serves as the basis of the floodplain ecosystem and the resource base for cattle breeding. The unflooded territory is the basis of the social subsystem and irrigated agriculture. An unstable flooded area has only recreational value, so a progressive increase in its area due to a decrease in the area of the territory rest is a sign of degradation.

An analysis of the dynamics of the aggregated complex territorial structure of the VAF in Figure 20 shows a trend towards a decrease in the share of a stable flooded territory in all its elements. The smallest decrease in this share occurred in ecological territories. This can be explained by the fact that catastrophically degrading areas of wetlands and water meadows make up a smaller part of the entire territory of the VAF that belongs to the ecological type.

Figure 16 demonstrates a slight change in all three criteria for the state of the VAF territory with a significant change in the area of the stably flooded territory, which is due to the low proportion of the area with high sensitivity to the stability of flooding. In addition, an analysis of the characteristic functions shows that their first and second pairs depend oppositely on the frequency of flooding. Therefore, most of the changes in these functions compensate each other with a decrease in the frequency of flooding of the significant part of the VAF.

It should be noted that the linear extrapolation over time of the degradation factor of the Volga River bed apparently leads to an overestimation of the effect of future degradation in the floodplain. However, our results give a five-fold reduction even for the upper estimate of the area of the steady flooded area of the VAF in 2050 compared to 1991. This means the destruction of the floodplain ecosystem and grassland management by 2050. A more accurate forecast of this effect requires a physically based model of the state dynamics of the Volga River and small interfluve channels.

## 6. Conclusions

The new approach to the analysis and forecasting of the state of the floodplain territory based on hydrodynamic and geoinformation modeling and statistical processing of natural observational data is proposed. The use of high-performance computing made it possible to build and analyze approximately a thousand digital flood maps for natural and virtual hydrographs. This made it possible to study the problem of degradation of the territory of

the northern part of the Volga-Akhtuba floodplain due to slow natural and human-made changes in the Volga riverbed.

The main tool of our analysis is the spatially distributed model of the complex structure of the floodplain territory, which was created as a superposition of the hydrological and socio-natural economic structures of the VAF. The existence of the distribution function of the volume of the annual spring flood with the invariance of the relief and the channel structure of the floodplain ensures the stability of its hydrological structure.

It is important to note that the studied hydrological structure of the floodplain area is very difficult to determine by field research. Therefore, we substantiate its existence by statistical methods. Indeed, we can determine the boundaries of areas flooded with one or another frequency using a set of flood maps of the floodplain over $T$ years. However, other $T$-year periods may give other boundaries. Therefore, it is necessary to find the minimum allowable value of $T$, which ensures the invariance of these boundaries with a given accuracy. Both the objectivity of our approach and the possibility of revealing the real variability of the found structure are determined by the size of the general population $N$, i.e., the duration of the observation period.

The elements of the hydrological structure are fragments of the territory with the same kind, which characterizes the frequency range of floods. The number of ranges and their sizes are determined by experts based on the purpose of modeling. Large ranges that combine several kinds are what we call types. We used three large frequency ranges in the study of the Volga-Akhtuba floodplain: stable flooding (with a frequency of more than 0.85), unflooding (with a frequency of 0) and unstable flooding (with a frequency of 0 to 0.85).

We propose to build a functional $\mathcal{F}$-structure of the floodplain territory on the basis of its cadastral map, which takes into account social, environmental and economic characteristics. The territory fragments of one kind of use are elements of this structure. We combine kinds of use into three types: social, economic and ecological. The elements of the complex structure of the floodplain territory are fragments of its territory, each of which belongs to one hydrological and one functional kind. The elements of the aggregated complex structure are territories of one hydrological and one functional type.

The state of the environmental and socio-economic system of the northern part territory of the Volga-Akhtuba floodplain is estimated by the values of social, natural and economic criteria. Their assessment on the set of kinds of the functional structure elements is based on the family of characteristic functions that determine the degree of correspondence between the hydrological kind of the element and its functional kind.

We used the results of observations of the level of the Volga River below the Volga Dam for 60 years to build a phenomenological model of the dynamics of morphological changes. We justified the use of data on the magnitude of spring floods over 30-year periods as an acceptable sample size when constructing their distribution function over 60-year observation period. This made it possible to achieve an acceptable accuracy in determining the elements of the $\mathcal{H}$-structure, to build both retrospective and modern $\mathcal{H}$-structures and to estimate the forecast accuracy of the structure and criteria for the territory state of the northern part of the VAF until 2050.

Our forecast made it possible to identify areas in the floodplain that may lose their flood stability in the coming decades. This poses the problem of its development, which can be solved either by changing its functional purpose, or by implementation of technical projects that preserve its hydrological type. The same applies to the territory, which has already lost the stability of flooding by now. On the other hand, the calculation of the hydrological structure of the floodplain territory of the past decades can serve as a guideline for restoring its environmental and socio-economic system and ensuring its sustainable development. The strategy for the sustainable development of such territories should include, first of all, hydrotechnical projects that neutralize the factors of technogenic channel degradation. Another part of this strategy should be projects that change the socio-economic purpose of areas that can change their hydrological properties.

Thus, our forecast can serve as the basis for designing an integrated system for managing the development of a territory. This system may include hydrotechnical projects for watering and hydrological safety, coordinated with projects for the socio-economic development of the territory. We also note the possibility of ecological and economic management of economic entities of the territory on the basis of the division of the their actions damage and the channel degradation damage. Finally, let us point out the prospects for designing a bypass canal from the Volgograd reservoir to the Akhtuba river bypassing the hydroelectric dam, which can replace the accelerating shortage of flood waters in the coming decades.

The approach presented here to assess the state of the floodplain, combined with hydrodynamic simulation to justify hydrotechnical projects for regulating the flood regime, can become the basis for the creation and operation of a sustainable development system for the territory of the Volga-Akhtuba floodplain.

**Author Contributions:** Conceptualization, A.A.V. and A.V.K.; methodology, I.I.I.; software, I.I.; validation, I.I.I. and M.A.K.; formal analysis, I.I.I. and A.V.K.; investigation, I.I.I. and A.A.V.; resources, A.V.K.; data curation, M.A.K.; writing—original draft preparation, I.I.I.; writing—review and editing, A.A.V. and A.V.K.; visualization, I.I.I.; supervision, A.A.V.; project administration, A.V.K.; funding acquisition, A.V.K. All authors have read and agreed to the published version of the manuscript.

**Funding:** This research was funded by the Ministry of Science and Higher Education of the Russian Federation (the government task no. 0633-2020-0003, A.A.V. and A.V.K.).

**Institutional Review Board Statement:** Not applicable.

**Informed Consent Statement:** Not applicable.

**Data Availability Statement:** All data used in this paper are taken from the open sources and the references are given.

**Acknowledgments:** The authors are grateful to Sergey Khrapov for providing the software for numerical simulation and to Anna Klikunova for the actual versions of the DEM and DHLM.

**Conflicts of Interest:** The authors declare no conflict of interest. The funders had no role in the design of the study; in the collection, analyses, or interpretation of data; in the writing of the manuscript, or in the decision to publish the results.

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
