# Peer review of "Modeling the Territorial Structure Dynamics of the Northern Part of the Volga-Akhtuba Floodplain"

_computation, doi:10.3390/computation10040062_

Round 1

Reviewer 1 Report

Paper presents an interesting and complex study, but some revisions such as follows are recommended:

  • in line 131, please revise "...a domed to a two-stage shape during the VHPS existence(Figure 2)." 
  • a reference to figure 6 is missing in the paper text;

Reviewer 2 Report

The paper proposes a new approach to analyze the floodplain territory and forecast. The topic is interesting, the methodology is appropriate, and the paper is well-written.

In what follows, some points that I think the Authors should fix:

  • Line 4: Authors mention the software ECOGIS, please, add a link to find it.
  • Line 5: The paper focus on the “territory of the Volga-Akhtuba”. Please, provide any geographical information on location of this territory.
  • Line 39: Authors mention the “The Yellow River and the Yangtze River floodplain “. Please, explain where they are located
  • PAG 4 – Figure 2 – panel c: Better explain the meaning of Theta_1 and Theta_2
  • PAG 4 – Figure 2 – panel c: is Q_1^m, what in the Q^m(t) of Line 203?
  • PAG 4 – Caption of Figure 2: Explain the meaning of Q_{VHPS}
  • Line 126: There is a missing space between “VHPS existence” and “(Figure 2)”
  • Overall comment on Section 1: This section is very long (about 5 pages) and only at the end there is brief idea of the aim of the paper. Please, add, preferably at the beginning, what is the aim of the paper and what kind of analysis you are going to apply.
  • PAGE 6 – Before Line 183: Please, better introduce your notations, especially for subscripts.
  • PAGE 6 – Before Line 183: Authors say that the DHLM has been proposed by Anna Klikunova. Please, add a reference.
  • Line 185: I think that the article “The” is not necessary before Figure 3. Please check along the paper (i.e. Line 196).
  • PAGE 7 – Figure 4: The caption of Figure 4 is not very clear. Please, better refers to the maps, for instace, by adding different letters as for Figure 2.
  • Line 214: Authors write: Q=Q_2^m < Q=Q_1^m. This inequality is not clear for me. Probably is better to denote differently, since you have Q<Q.
  • PAGE 9 - Equations 2-3-4: I think that some previous notation is required to understand these equations.
  • PAGE 9 – f^(turb): This equation is not very clear since it is written in a recursive way. Please check.
  • PAGE 10 – Equation 6: g is not defined.
  • PAGE 11: After equation (8), Authors define rho_a. I think, that you have to anticipate that equation since the parameter appears before, at line 244.
  • Line 257: The values of q_b (q_b>0 and q_b<0) is not clear.
  • Line 325 and Line 339: I think that the unit should be added to 0.2.
  • Line 347: What does “large enough” mean?
  • PAGE 14 – Equation 12: Since you define the mean and the standard deviation by means of a parameter “r”, I think that that parameter should appear in equation 12.
  • PAGE 15 – Equation 13: in mu_2 definition there is a comma. Please, remove it.
  • Line 390: Authors affirm “we generate k corresponding sets of..” Please, specify the values of k or at least a plausible range.
  • PAG 16: It seems that Table 1 is not cited in the text
  • Line 434: Along the paper, you refer to all figures as Figure, here you write Fig. Please, uniform the way to refer to figures.
  • Line 518: Please, check the sentence (English).

Reviewer 3 Report

EVALUATION

In the document, the authors describe an approach to analyze the dynamics of the territorial structure of floodplain territories of regulated rivers based on statistical processing of observational data, hydrodynamic modeling using high-performance computing. The unit of analysis is located on the territory of the Volga-Akhtuba floodplain.

I think it is a relevant, current topic and interesting reading for researchers, students, and professionals working or interested in the area, because, in recent years, publications related to geographic information have gained strength in several areas of knowledge. Since geographic information systems are powerful tools that allow, store, consult, analyze and edit information related to a geographic space.

The document is formalized with the following sections: 1. Introduction; 2. Numerical modeling of the hydrological regime of the floodplain area; 3. Modeling the floodplain territorial structure: methods, instruments and technology research; 4. Results of modeling the dynamics of the ESES territorial structure for the Northern part of the Volga-Akhtuba floodplain; 5. Discussion; and 6. Conclusions. I think they are enough for the document. However, I understand that the document is very dense and could have some revised sections.

Next, I will try to make some comments and suggestions for the sections that make up the document.

I am of the opinion that the “Introduction” aims to: explain the general problem; define the research problem; present the background on which the study is based (bibliographic review) and define the objectives of the study. I think that the authors identified the problem, competently described it, and highlighted its magnitude and importance, however, in my view, the purpose of the research was not made explicit. I think that although it presents the objective of the study, it does not give due emphasis to its importance and scope, as well as its limitations.

In section “2. Numerical modeling of the hydrological regime of the floodplain area” the authors present the main components of the computational model of the hydrological system, as well as contextualize its application within the focus of the study. A point that I consider important is that according to the authors “The technological basis of our study is a set of algorithms and ECOGIS software”. ECOGIS is a tool for consulting ecological flows, among other features. The use of tools such as ECOGIS makes it possible for water resource planners to make queries, generate reports and create graphs of stored information about ecological and circulating flows. Similar to the one presented in the document. However, the software is not addressed or presented in the document. I think this part should have been better explored. Perhaps even in section 3, it refers to “Modeling the floodplain territorial structure: methods, instruments and technology research”.

Section 4 deals with the Results of modeling the dynamics of the ESES territorial structure for the Northern part of the Volga-Akhtuba floodplain. I think the results are consistent with the method indicated above.

The epilogue of a survey is to show the results. The results are supported by eleven figures, which seek to make this part of the document more pleasant to read and understand.

Regarding section “5. Discussion”, I think that the authors commented on the relevant results of the study carried out, although they could have better highlighted the new and important aspects. I think it would also be important to compare the results obtained with those of similar works by other authors.

For section “6. Conclusions”, I would suggest emphasizing the new and important aspects of the study and the conclusions that are derived from them. I remember that people usually read the abstract, the introduction, and the conclusion.

The following are some specific comments for the other parts of the document:

Title: is of adequate length, with 13 words, (Modeling the territorial structure dynamics of the Northern part of the Volga-Akhtuba floodplain). It should be borne in mind that the title is the first component to be read in an article and, therefore, it is the most important sentence of the article. I think the text presented answers the three fundamental questions: 1. What was done? 2. What was done about it? 3. Where was it made?

Abstract: It is very important due to its significant use in electronic databases. The text presented in the document is of adequate size with 273 words, distributed in 9 sentences and presenting an average of 30.33 words/sentence. I think that the average number of words used in the sentences is very high, and it can be difficult to read, for those who are not very familiar with the subject. For a better reader experience, it is recommended to write more objective sentences. According to the Oxford Guide for Writing (2020):

  • Sentences of up to 12 words are easy;
  • Sentences of 13-17 words are acceptable;
  • Sentences with 18-25 words are difficult;
  • Sentences longer than 25 words are very difficult.

Regarding construction, I think that the abstract presents structure problems and, in my opinion, does not present the necessary items for this part of the document. Namely: “What was studied” (Introduction); “How the study was carried out” (Materials and Methods); “What was found” (Results) and “What it means” (Conclusion). I think the text presented does not meet these requirements. I suggest it be rewritten.

Another pertinent observation is that the abstract must be written in the past tense, except for the last paragraph or the concluding sentence.

Keywords: the authors present 5 (hydrodynamic and geoinformation modeling; high-performance computing; environmental and socio-economic system; territorial structure; Volga-Akhtuba floodplain). I think that a number is adequate. I think it is desirable that the order of presentation of keywords is from the most comprehensive to the most specific, which is not the case in the document.

References: The authors provide a list of 78 references that are all cited in the document. I think that the number of references is adequate, for the profile of the proposed, and consists of 25 pages. Regarding the timeliness of the references, 53 (67.9%) have five or less, that is, 32.1% are more than five years old, and 10.25% are more than ten years old. I think it's a good percentage for the document profile.

Graphic Elements: Figures, tables, and charts are intended to communicate information visually and quickly. The authors present 1 table and 21 figures, which I think is a good number for the document. They are properly identified and their captions are understandable. I think that to improve the understanding of the document some should be better cited in the text. Another problem I've found is that some are excessively large and extend beyond the page boundaries and would need to be adjusted.
